# Throughput Optimization of Multichannel Allocation Mechanism under Interference Constraint for Hybrid Overlay/underlay Cognitive Radio Networks with Energy Harvesting

**Hakan Murat Karaca**

Department of Computer Engineering, Celal Bayar University, Muradiye-Manisa 45030, Turkey;
hakan.karaca@cbu.edu.tr; Tel.: +90-236-201-2113

**Abstract:** By harvesting energy from ambient radio frequency (RF) signals, significant progress has been achieved in wireless networks self-maintaining their life cycles. Motivated by this and improved spectrum reuse by combined use of overlay/underlay modes of cognitive radio networks (CRNs), this paper proposes a novel multi-channel (m-channel) allocation performance maximization algorithm for low-power mobiles. CRNs, called secondary transmitters (STs), can harvest energy from RF signals by nearby active primary transmitters (PTs). In the proposed scheme, PTs and STs are distributed as independent homogeneous Poisson point processes and contact their receivers at fixed distances. Each PT contains a guard zone to protect its intended receiver from ST interference, and provides RF energy to STs located in its harvesting zone. Prioritization of STs during opportunistic allocation of channels is critical as properties like energy level and harvesting capability improve channel distribution performance. A novel metric is proposed that prioritizes STs based on initial energy levels, harvesting capability, and number of channels through which they can transmit. For comparison, three algorithms were considered: a greedy mechanism for m-channel allocation of hybrid CRNs without harvesting, the proposed m-channel allocation schemes based on maximum independent sets (MIS), and the proposed metric of hybrid CRNs with harvesting capability. The simulations show that the proposed m-channel allocation method based on MIS outperforms the greedy algorithm. The proposed m-channel allocation using the proposed metric on hybrid CRNs with energy harvesting ability produced the best performance of the three methods, proving the superiority of the proposed algorithm.

**Keywords:** cognitive radio networks; energy harvesting; hybrid underlay/overlay scheme; maximum independent set; multi-channel allocation; interference constraint

## 1. Introduction

Energy requirement is an important issue in wireless communications [1]. The service life of wireless sensors dramatically affects the reliability and performance of wireless networks. To address this issue, different methods have been devised, with the main goal of improving the energy efficiency of wireless sensors [1]. Beside reducing energy consumption, recent studies aimed to harvest energy from radio frequency (RF) signals to mitigate the problem of energy shortages. Energy harvesting (EH) is used to convert RF signals into energy, which can then be used for other objectives like data processing and transmission. Mobile devices work with energy supply from a battery, which needs to be regularly physically charged or replaced.

Operating mobile devices with energy sources, like solar, acoustic, vibration, thermal, wind, and even ambient radio signals, has been investigated with the aim of reducing energy costs or

potentially harmful effects to the environment [2]. Specifically, energy harvesting technology from RF signals has been designed to supply energy to wireless nodes. RF energy harvesting provides relatively more predictable energy compared with other energy harvesting techniques. The amount of RF energy that can be harvested depends on the distance between an RF energy source and the harvesting device and wavelength of the harvested RF signal [3].

Spectrum demand has increased with the rapid increase in usage of both wireless devices and wireless services. The Federal Communications Commission (FCC) reported that the spectrum is not highly used [4], specifically in wireless sensor networks [5]. As a consequence, cognitive radio technology has emerged to enable spectrum reuse [6]. In a cognitive radio network (CRN), secondary users (SUs) can opportunistically access the spectrum allocated to the primary users (PUs), who are the licensed owner of the spectrum. To mitigate the energy shortage problem, green communication is a promising approach that has recently received increased attention [7]. Supplying renewable and clean energy sources to the network nodes that decreases energy consumption and effects on the environment is critical [2].

Usage of energy harvesting (EH) in a CRN involves both energy and spectral efficiencies. The energy harvesting capability in a CR improves energy efficiency, enabling it to work without a wired external power supply. In a CRN, sensing an idle spectrum is a critical task to ensure PUs are not disturbed by SUs with a wrong detection decision [8]. Generally, mechanisms used for spectrum sensing are based on energy detection [9,10]. The sensitivity of RF EH is improving, so EH is expected to be effectively and widely used for spectrum sensing in the near future, even though spectrum sensing and RF EH have different power sensitivity levels, for instance, $-60$ dBm for spectrum sensing [11] but $-10$ to $-30$ dBm for RF EH [12]. This means SUs are expected to collect energy from the RF signals transmitted by PUs after detecting whether the channel is occupied. Wireless devices like SUs can harvest energy from RF signals if they are equipped with RF harvesting capability and then can use this energy for sensing and transmission. If the source of RF transmission (like PUs) is near the harvesting devices, signals can be collected and turned into direct current (DC) electricity. After that, harvesting energy can be stored and then consumed to achieve sensing and data transmission. SUs can transmit data when they are out of the PU interference range or even inside the interference range if the channel is found to be idle. However, to create harvesting energy, SUs should sense both idle and busy channels. Considering the trade-off between spectrum efficiency and RF energy supply, energy harvesting seems to be a promising solution for CRNs [13], and various studies [10,14–20] have contributed to the throughput [10,14–16], outage probability [19,20], and sensing threshold [17,18].

In this context, many studies examined harvesting in CRNs [14,17,21–25]. Park et al. [17] proposed an efficient spectrum sensing strategy to maximize the total throughput of SUs under energy and collision constraints. Hoang et al. [21] proposed an effective channel access strategy for SUs based on the Markov decision process (MDP). In addition, Zheng et al. [22,23] studied the combined information and energy cooperation strategy between PUs and SUs. Pratibha et al. [24] updated the homogeneous Poisson point processes (HPPPs) to construct a decentralized channel-selection mechanism to improve the throughput of SUs in multi-band CRNs. Bae et al. [25] examined how sensing probability, access probability, and energy queue capacity affect the manageable throughput of a multi-user CRN.

Lee et al. [10–14] showed that energy from ambient radio frequency (RF) from primary transmitters can be harvested by SUs, and the transmission probability and final throughput under the outage constraint were found. Yin et al. [15] investigated the trade-off between harvesting and sensing throughput, and the expected manageable secondary throughput was improved while protecting primary users. Park et al. [11,12] focused on the trade-off between spectrum sharing and sensing; to maximize the achievable throughput, they made optimization on the detection threshold of spectrum sensor of the SU.

In another article [26], RF EH-enabled cognitive radio sensor networks were investigated under an energy causality constraint, which forces the total consumed energy to be less than the total harvested

energy. The authors suggested an optimal mode selection strategy that tries to create an equilibration between the harvested RF energy and immediate throughput in harvesting and transmitting modes.

Lee et al. [14] represented the mobile equipment in a secondary network opportunistically by either harvesting RF energy from transmissions of nearby devices in a primary network, or transmitting data if outside of the interference range of any other primary network. In addition, the equipment maximizes the throughput of the secondary network by proposing the optimal transmit power and number of the secondary transmitters under an outage probability constraint. Barroca et al. [27] analyzed a cognitive wireless body area network with RF energy harvesting capability and addressed challenges with the physical, medium access control (MAC), and network layers and discussed some possible solutions. They proposed some useful solutions for cognitive radio-enabled RF energy harvesting equipment for combined data reception and RF energy harvesting. However, the issue of dynamic spectrum access for RF-powered CRNs has not been fully studied.

The technology used to maximize the spectrum use of CRNs can be approximately classified into overlay, interweave, and underlay modes [28–32]. In overlay mode, the free licensed spectrum was egoistically used by the secondary transmitter (ST). In underlay mode, ST simultaneously exists with the primary transmitter (PT) egoistically if the interference caused by the ST to primary receiver (PR) does not exceed a certain threshold [29] to satisfy quality of service requirement. They proposed a hybrid CRN model by which they maximized SU throughput under the partially observable Markov decision process framework. In Senthuran et al. [28], the SU toggles its transmission mode between overlay and underlay, and uses the latest-sensed data for channel selection. Ma et al. [30] defined the subchannel allocation problem as a coalition formation game in hybrid overlay/underlay cognitive femtocell networks, and proposed an updated recursive core algorithm to improve the total network throughput.

Lee and Zhang [31] studied underlay/overlay-based spectrum sharing models of cognitive wireless powered communication networks (CWPCNs) and maximized the total throughput by altering its transmission mode. Kim et al. [32] expanded a previous analysis [31] to multi-input multi-output CWPCNs, through which several impressions on multiple antenna gains were received.

This study focused on optimal m-channel allocation under both the energy and collision constraints using the hybrid overlay/underlay mechanism, where spectrum reuse was increased by enabling energy harvesting without affecting the primary transmission. The energy constraint guarantees that the harvested energy should exceed the consumed energy, and ST should decrease spectrum usage attempts in cases where there are energy shortage situations [33]. For the collision constraint to be satisfied, the interference caused by STs on primary transmitters (PTs) should be over a certain signal to interference plus noise ratio. ST harvests energy from available primary users' signals, like energy-harvesting CRNs in [29]. On the other hand, in terms of interference caused by PTs on secondary receivers, which mainly depends on distance between them, the aim of the secondary receiver is to be able to decode the primary signal only to help to accomplish a better secondary rate. However, it is not the scope of this paper to analyse this effect. In the literature, there are several studies on this topic. For instance, opportunistic interference cancellation (OIC) mechanism proposed in [34] can be utilized for this purpose by (a) selection of the data rate in the PT and (b) the link quality between the PT and the secondary receiver.

However, in this journal, the ST is assumed to have two transmission modes: overlay and underlay. The transmission mode is determined according to its distance to the primary transmitter, remaining energy in the battery, and result of spectrum sensing activity [33]. Accordingly, the circular transmission area of a primary transmitter is divided into overlay mode area, underlay mode area, guard zone, and harvesting zone (HZ). As seen in Figure 1, the overlay mode area starts where the coverage area of PT ends, the underlay area is defined between the end of guard zone and end of the coverage area, and the guard zone is the area where no ST can transmit to protect its intended receiver from ST interference. The HZ is the area where an ST can harvest radio signals of an active PT as harvesting is only possible for a certain distance due to attenuation.

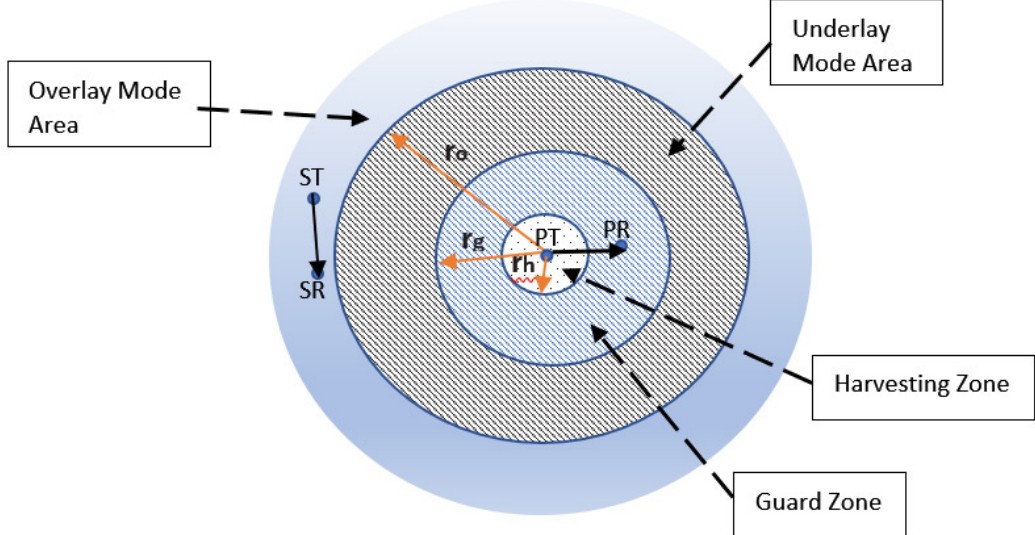

**Figure 1.** Allocation of coverage area into different zones. ST = secondary transmitter; SR = secondary receiver; PR = primary receiver; PT = primary transmitter.

In order to determine in which area an ST is located, the mobility model of an ST is not specified as the statistical information of ST locations is only needed in the analysis part of this study. Random walk mobility model [35] may be used to obtain statistical information of locations. Based on the statistical information of ST locations, the probability that an ST exists in any of areas can be determined. However, it is not the main focus of this paper which approach to use to obtain these probabilities. Therefore, these probabilities are obtained by open approaches based on the mobility model, the area in each zone, and the initial probability distribution of ST. ST knows the area where it locates by the received power of RF signals and the interference threshold of primary users.

In this journal, a hybrid m-channel allocation mechanism is proposed that maximizes the m-channel allocation efficiency in EH-based CRNs, where a ST is capable of harvesting energy from the RF signals transmitted by a PT. Different from a previous work in [33], a multi-channel allocation mechanism was investigated for hybrid CRNs by dividing coverage area into four areas, including a guard zone to protect the primary transmitter. In addition, when allocating channels, STs are prioritized by proposing a new metric considering energy levels, number of ST transmission channels, and harvesting capability, which improves and speeds up the allocation of channels. To the best of my knowledge, this is the first attempt to maximize m-channel allocation performance in a hybrid CRN with energy harvesting capability. The main contributions of this article can be summarized as follows:

(1) For the CRN with mobile energy harvesting ST, this paper is the first attempt to maximize the efficiency of m-channel allocation by dividing the coverage area into overlay mode area, underlay mode area, and harvesting and guard zones for the throughput analysis given the hybrid underlay-overlay scheme under the interference constraint.

(2) For allocation of multiple channels, to prioritize STs, a new metric for each of ST is proposed based on their current energy level, number of channels they need (having more channels means having higher priority), and harvesting capability to assign channels. For the energy constraint, at every time slot, the energy level of each ST is controlled; if exceeds sensing + transmission energy, it performs sensing and decides to transmit based on whether idle channels are found. Nodes lying outside of coverage area of a PT channel owner can use the channel simultaneously with the channel owner. As a result, interference is assumed to be negligible. Therefore, this ST does not need to sense the channel, so energy is only consumed for transmission, producing considerable energy savings.

(3)　Simulations show that channel distribution performance is markedly improved by harvesting energy from ambient RF signals when assigning channels to SUs for both the proposed new metric and the maximum independent sets (MIS) of the graph when hybrid zone partitioning is applied. In addition, as harvesting depends on number of users located in the defined HZ (density of SUs), the more users located in the HZ, the more energy can be harvested, which enables the storage of more energy, thereby increasing channel distribution performance.

The robustness against signal to interference and noise ratio (SINR) was also examined and found that it decreases with increasing SINR (outage) threshold as expected. The performance considerably depends on the Poisson distribution density of SUs. To the best of our knowledge, this is the first study to investigate the effect of harvesting on m-channel allocation performance for hybrid cognitive radio networks under the interference constraint. The remainder of this paper is organized as follows. Section 2 introduces the system model and notations, as well as explains the greedy mechanism and proposed schemes. Simulation results and their discussions are provided in Section 3. Section 4 provides the concluding remarks of this paper.

## 2. Materials and Methods

### 2.1. System Model

The allocation of the spectrum among all CR users based on their usage is called spectrum sharing. Accessing the channel so that the PU is protected is one of the spectrum sharing responsibilities. In general, spectrum sharing based on access technology is divided into three categories: overlay, underlay and interweave access techniques [36,37]. In the interweave access technique, a spectrum hole is available for the SU and it is allowed to use it without restrictions. If the PU returns to the spectrum, the SU has to leave the spectrum or change to another free channel. Therefore, the most important functionality of the SU is to sense the spectrum holes (holes are part of the spectrum band assigned to the PU, but they have not been used at a specific time and place). Spectrum sensing is the capability of an SU to sense the spectrum based on common detection methods like coinciding filter and energy detector. For overlay and underlay schemes, a SU can colocate with the PU without interference or with a minimum interference. In the underlay access technique, if the SU adjusts its transmission rate to prevent destruction of the PU, the spectrum efficiency increases. Adjusting the transmission power with minimum interference level, as well as using the channel capacity, to be efficient for both PUs and SUs, are the most critical limitations of this technique. In overlay mode, the SU can transmit at the same time as the PU. The interference resulting from the ST to the PR can be compensated by using portion of the SU's power to relay the PU's message. The SU functions as a relay to a PU and in turn the PU permits it to utilize a portion of its spectrum. Powering a cognitive radio network (CRN) with RF energy can provide both a spectrum- and energy-efficient solution for wireless networking, where harvested energy can be used for both data transmission and channel sensing, which both increase channel allocation performance. So, cognitive devices must not only identify spectrum holes for opportunistic data transmission, but also search for occupied spectrum bands to harvest RF energy.

Combining these requirements and issues, the goal of this study is to investigate the impact of RF energy harvesting on the m-channel allocation mechanism of a hybrid cognitive radio (CR) type of network and to improve channel distribution using the proposed metric and MIS-based allocation over greedy allocation mechanism. In the proposed model, static primary and secondary pairs coexist and are distributed as independent homogeneous Poisson point processes (HPPPs) in a unit square area, and the shape of the area does not affect the results. As shown in Figure 1, the unit circular coverage area of a PT is divided into overlay mode area, underlay mode area, guard zone, and HZ.

In this journal, guard zone is defined as the area where the ST cannot use the occupied channel of the PT, even if the power level is minimized. The radius of the guard zone is given as $r_g$ to protect its intended receiver from ST interference, and simultaneously delivers RF energy to STs located in its HZ.

To use the energy from ambient RF signals, the energy harvester in the ST should be equipped with a power conversion circuit that can extract DC power from the received RF signals [14,38]. Such circuits have practical sensitivity requirements, i.e., the input power needs to be larger than a predesigned threshold [14]. HZ is defined by the radius of $r_h$. To use multiple channels, each ST is assumed to have more than one wireless interface and energy harvester.

There are a PT and a corresponding primary receiver (PR), which both constitute the primary pair. The PT exists at the center of the unit area, whereas the PR is located far away from the PT, and receives primary packet transmissions. The spectrum licensed to the primary network toggles its state between idle and occupied. The secondary pair contain an energy-harvesting ST and a secondary receiver (SR) and they are randomly Poisson distributed as mentioned above. This study supposed node locations were static, and there existed a centralized server in the CR network. SUs broadcast their locations, available channels, and energy levels to that server, which is called the spectrum server, as well. Therefore, flow routing and spectrum/energy management are straightforward and organized. During flow of algorithm, SUs need to communicate with spectrum server, and the center should inform about the energy level/allocation results to users. It is supposed the interaction between SUs and spectrum server does not have any impact on spectral efficiency (communication occurs in a band outside of hired spectrum [39]). Each ST can be in one of the following three modes at any given instant:

(1) harvesting mode if it is inside the HZ of an active PT and does not have enough energy for transmission;
(2) overlay transmitting mode if it has enough energy and is located outside of the coverage area of an active PT, or anywhere if the PT is inactive;
(3) underlay transmitting mode if it has enough energy and is located inside the coverage area of an active PT by reducing transmitting power; and
(4) idle mode if it is located inside the guard zone, or is neither fully charged nor inside any of the HZs.

The ST transmission power is supposed to be small enough to satisfy the low-power requirement of RF energy harvesting. Overlay mode area consists of a disk at whose center the PT is located with radius $r_o$. The radius $r_o$ is defined according to the interference threshold of PR and the transmission power of the ST in underlay mode. Let $P_u$ show the transmission power of SU in underlay mode, and $P_i$ represent the interference threshold of PR. We have $P_u \times r_o^{-\alpha} = P_i$, where $\alpha > 2$ is defined as the path-loss exponent.

Instinctively, the PR should be located at the center of the overlay mode area with radius $r_o$, however is defined as above to make the analysis easier [40]. In the overlay mode area, energy cannot be harvested from the RF signals, and ST transmits in overlay mode when the licensed spectrum is sensed as free, but it is not permitted to transmit in underlay mode when the licensed spectrum is found busy because of the high level secondary interference of underlay mode transmission to the PR **unless transmission power level is decreased**.

The underlay mode area is defined as the area region outside of guard zone and inside of region with radius $r_o$, which means area where radius satisfies $r_g < r < r_o$, as shown in Figure 1. In the underlay mode area, it is possible to tolerate the secondary interference of underlay mode transmission to PR by decreasing transmission power of ST. Based on the distance between ST and PR, power level is reduced such that interference level can be tolerated to satisfy required SINR. However, outside of region with radius $r_o$, interference level becomes ignorable (which corresponds to interference region of PT). After this point, if the residual energy is sufficient and the licensed spectrum is tested as busy, ST may transmit without the need to decrease power level, which implies overlay mode transmission, that is why the region is called here overlay mode area. On the other hand, if the licensed spectrum is tested as free and the residual energy is enough, then the overlay mode transmission may occur anywhere.

In order to prevent collisions with the primary pair, ST needs to sense spectrum holes properly [41]. The secondary pair synchronizes with the primary pair, and ST senses spectrum based on the residual energy in the battery. However, if the ST lies outside of interference range of a PT, it can use that channel even if it is busy, so there is no need to sense channels, which saves the energy required for sensing for that ST. If the sensing period is defined as $\tau$, and time slot is T, transmission period is given as T – $\tau$, which is explained in Figure 2 [42].

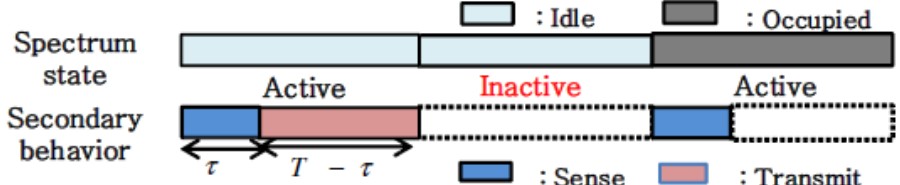

**Figure 2.** Opportunistic spectrum access of energy-harvesting cognitive radio (CR). The ST harvests energy, $EL_T^h$ , and consumes $EL_T^c$ , for opportunistic spectrum access with sensing duration, $\tau$, and slot duration, T.

The secondary transmitter harvests energy, $EL_T^h$, and consumes it ($EL_T^c$) for opportunistic spectrum access with sensing duration $\tau$ and slot duration, T. Due to the randomness of energy arrival, the ST chooses either active mode or inactive mode based on the residual energy, $EL_t$ at the beginning of time slot t.

Since the aim is to maximize secondary throughput, it is supposed that ST always has packets to transmit. Let $s_t \epsilon$ 0 (not access), 1 (access) depict whether ST can make spectrum sensing at the start of time slot t. It indicates if the residual energy $EL_t$ is sufficient for spectrum sensing and that of overlay mode transmission. Let $r_t \epsilon$ 0 (idle) and 1 (occupied) show the result of spectrum sensing at time slot. So, let us define a greedy spectrum access mechanism as:

$$s_t = \begin{cases} 0, & EL_t < EL_s + EL_o \\ 1, & EL_t \geq EL_s + EL_o, \end{cases} \tag{1}$$

where $EL_t$ represents the resulting energy at the beginning of time slot, and $EL_s$ shows the energy used when spectrum sensing. Let $EL_o$ and $EL_u$ denote the energies consumed by the overlay and underlay mode transmissions in one time slot, respectively. In underlay mode area, as ST is located close to PT, power level should be low in order to protect PT from excessive interference. However, in overlay transmission mode, either channel is tested as free or ST is located outside of interference region of PT, so there is no harm to increase the tranmission power level of ST until the PT falls within its interference range. As a result, there is no drawback to suppose $EL_o > EL_u > 0$. When $s_t = 1$, ST senses spectrum and finds whether the channel is occupied or not.

When sensing period, transmission period, sensing power, and ST overlay and underlay transmission powers are given, energy levels can be calculated by:

$$EL = P \times T, \tag{2}$$

where $EL$ is energy level in Joule, $P$ is power level in W, and $T$ is duration in seconds when power level $P$ is used. Following scenarios can be considered according to the location of the ST, remaining energy, and licensed spectrum state:

- ST decides whether to sense spectrum based on the residual energy in the battery.
- If the ST cannot make sensing and is located in any HZ of an active PT, the ST harvests energy from RF signals or stays idle if it is located in the underlay, overlay mode area, or in the guard zone.
- If the ST can make spectrum sensing, it does so, and the result is obtained.

- When the licensed spectrum is found as free, ST makes transmission in overlay mode. So, the resulting energy at the end of time slot t is: $EL_{t+1} = EL_t - EL_s - EL_o$
- When the licensed spectrum is tested and found to be busy, ST harvests energy from RF signals if it is located in the HZ, transmits in overlay mode if it is located outside of interference range, or transmits in underlay mode if it is located in the underlay mode area.

$$EL_{t+1} = EL_t + EL_h \text{(harvesting zone),} \tag{3}$$

$$EL_{t+1} = EL_t - EL_o, \tag{4}$$

(overlay mode, no need to sense as ST is located outside of interference region)

$$EL_{t+1} = EL_t - EL_s - EL_u \text{ (underlay mode).} \tag{5}$$

Prioritization of STs based on their energy level, whether they can harvest energy and number of channels they need are critical during distribution of channels to STs.

- Having higher initial energy enables an ST to sense channels and transmit data over longer time slots as their energy will be wasted in a longer period.
- If an ST is located in HZ of a PT, which means it can harvest energy, it will have more energy on the fly which means more chance to sense channels and make transmission.
- Having more than one wireless interface for transmission, meaning it has multi-channel usage ability, enables an ST to make transmission over more channels at the same time, which improves throughput of the total system because aim of a cognitive system is to maximize channel utilization.

So, from two nodes competing, the one with higher energy should obtain the channel. Nodes located in the HZ should be prioritized since harvesting capability contributes to have more energy, hence having more ability to transmit data. Additionally, multi-channel transmission capability of an ST should have higher prioritization during channel allocation.

Considering the above, this journal proposes a novel metric called PriorityMeasure below to define the priority for each secondary transmitter $ST(i)$ towards a channel $j$. That is why one of the proposed algorithms is based on this for prioritizing STs to assign channels.

$$PM(i,j) = ST_g(i,j) \times [0.4 \times EL(i)/max(EL) + 0.3 \times CN(i)] + 0.3 \times ST_h(i,j), \tag{6}$$

where

- $PM(i,j)$ denotes PriorityMeasure of $ST(i)$ for channel j;
- $ST_g(i,j)$ shows whether $ST(i)$ is located in the guard zone of $PT(j)$ (owner of channel $j$), which is 1 if located outside the guard zone and 0 otherwise as $ST(i)$ can never transmit using channel $j$;
- $EL(i)$ is the current energy level of $ST(i)$ in joules and $max(EL)$ is maximum energy level of all *STs*. So, component of the equation regarding energy level is normalized by maximum energy;
- CN(*i*) indicates the number of channels needed by $ST(i)$. Therefore, for instance, if number of channels required is 3, the contribution to PriorityMeasure will be 0.9; if 1, it will be 0.3. This component is appended to the metric, because, as mentioned before, contribution of *STs* owning higher number of channels on PriorityMeasure should be higher; and
- finally, $ST_h(i,j)$ is 1 if $ST(i)$ is located in the HZ of the $PT(j)$, and 0 otherwise. As an *ST* that can harvest should be located in guard zone by definition, this coefficient is separately added to the PriorityMeasure.

Coefficients 0.4, 0.3, and 0.3 are selected to prioritize and normalize each component in the equation such that sum of them becomes 1. The highest priority is given to the component representing

energy level as having higher energy is the most important factor for an ST to be able to make transmission.

The ChannelNeed of each ST is a random value uniformly distributed between 0 and $n$ (positive integer); initial energy levels are also uniformly random defined between 0 and Max(EL) (positive integer). In the mode considered here, the HZ of a PT is defined by a circle with a 33m radius, meaning a power level of 0.00126 mW ($\sim -29$ dBm) can be harvested, as explained previously [43]. Here, the guard zone is assumed as 75 m.

Primary transmitter channels are randomly defined as either idle or busy.

### 2.2. Greedy and Proposed Algorithms

In this study, a greedy mechanism is taken as a basis for channel allocation to compare the proposed mechanisms and highlight its performance. The results of the proposed MIS-based allocation and the allocation based on the proposed metric are also compared. The aim of all these mechanisms is to improve multi-channel allocation performance when utilizing energy harvesting.

#### 2.2.1. Greedy Algorithm

The input and output parameters and steps of the greedy allocation algorithm are given in Table 1.

#### 2.2.2. Proposed Algorithm Using MIS of Interference Graph

This algorithm's allocation mechanism is based on the method proposed previously [44]. However, energy harvesting mechanism, zone definitions, and energy constraints are added to the collusion constraint to harvest energy and improve channel distribution performance. The input and output parameters and steps of this algorithm are given in Table 2.

#### 2.2.3. Proposed Algorithm Using Proposed Metric for Allocation:

Proposed metric given in Equation (6) is used in the proposed algorithm. The input and output parameters and steps of the proposed algorithm are given in Table 3.

## 3. Results and Discussion

The numerical results used to evaluate the performance of the proposed mechanisms for energy harvesting CR are presented below. The system parameter definitions and corresponding values used in simulations are summarized in Table 4 [9,29]. The sensing power was set to 110 mW [45]. Energy levels to make overlay and underlay transmissions include energy needed for both sensing and transmission. So, if sensing and transmission durations are 0.002 ms and 0.098 ms, and sensing, overlay, and underlay transmit powers are 110 mW, 50 mW, and 30 mW, respectively, then energy levels $EL_o$, $EL_u$, and $EL_s$ are calculated according to Equation (2), which is given in Table 4. Harvesting conversion efficiency is considered to be as 0.75 [46,47]. Energy levels are given in mJ.

The model used in simulations consists of primary and secondary users which are distributed as independent HPPPs in a (1000 × 1000) m$^2$ area. Each primary user is supposed to have one channel, so the number of available channels equals the number of primary transmitters. Each secondary user represents an access point (base station or Internet of Things (IoT) sensor), which is assumed to be equipped with more than one wireless interface, so each secondary user may need more than one channel based on the number of wireless interfaces equipped. The unused spectrum from primary users form a spectrum pool with orthogonal frequency channels. A typical urban path loss model was adopted with a path loss exponent of 3.7.

All simulations were run with MATLAB (FIGES A.S, Istanbul, Turkey). To understand the implications of different SINR values, Figure 3 depicts information about signal strengths for 3GPP Long-term evolution technology (LTE).

| SINR | Signal strength | Description |
|---|---|---|
| >= 20 dB | Excellent | Strong signal with maximum data speeds |
| 13 dB to 20 dB | Good | Strong signal with good data speeds |
| 0 dB to 13 dB | Fair to poor | Reliable data speeds may be attained, but marginal data with drop-outs is possible. When this value gets close to 0, performance will drop drastically |
| <= 0 dB | No signal | Disconnection |

**Figure 3.** Descriptions of signal strengths for different signal to interference and noise ratio (SINR) values for Long-term evolution technology (LTE).

To demonstrate the performance and supremacy of the proposed algorithms, the channel distribution fairness (performance) for each of the mechanisms is compared. In order to show fairness of proposed allocation methods in this journal, the metric given in Equation (9) is used as the performance criterion [44], which also considers if any outage of PRs occurs due to exceeding the limit of interference threshold for the defined SINR. SINR is calculated as

$$SINR = (P_{P_T}/(d_{(P_T,P_R)})^{\alpha})/(No + \sum_{i=1}^{S} P_{C_i}/(d_{(C_i,P_R)})^{\alpha}) \geq \beta_{P_R}, \tag{7}$$

where

- $P_{P_T}$ is the transmission power of primary transmitter $P_T$;
- $d_{(P_T,P_R)}$ is the distance between primary transmitter $P_T$ and primary receiver $P_R$;
- $\beta_{P_R}$ is the tolerable SINR ratio threshold at $P_R$ (SINR value when total interference is $P_{U_{th}}$ which is the tolerable interference threshold at the primary receiver $P_R$), so that the corresponding primary receiver can successfully receive primary transmitter packets;
- $S$ is the total number of secondary users;
- $P_{C_i}$ is the transmission power of secondary user $s_i$;
- $d_{(C_i,P_R)}$ is the distance between secondary user $s_i$ and the primary receiver $P_R$; and
- $No$ is additive white Gaussian noise for the receiver, with a zero mean.

This is equivalent to guaranteeing a minimum rate for the primary receiver. Because more than one cognitive transmitter is using the same channel at any given time, the total interference is the sum of all interferences at the point of the receiver. For the SINR value to be less than $\beta_{P_R}$, the total interference should satisfy [44]:

$$\sum_{i=1}^{S} P_{C_i}/(d_{C_i,P_R})^{\alpha} < P_{U_{th}}. \tag{8}$$

The purpose of multi-channel allocation is to assign channels such that all STs utilize channels they need. To measure the fairness, in other words effectiveness, of proposed schemes, Equation (9) is proposed in [44]. For this purpose, the same metric is used to prove the fairness of the proposed schemes in this journal, as well. As more STs make use of channels, ChDistFairness gets smaller, which implies better allocation performance, and when all SUs get all channels they need, it becomes zero which means the best allocation scheme. That is why all proposed mechanisms aim to minimize ChDistFairness values.

$$\texttt{ChDistFairness}_{Gre} = \sum(Ch_{Need} - Ch_{Obt})/(Ch_{Need} \times S) \times$$

$$(1 - Prob_{Out}(\beta_{P_R}) + 1/S \times Prob_{Out}(\beta_{P_R}), \tag{9}$$

where ChDistFairness$_{Gre}$, Ch$_{Need}$, and Ch$_{Obt}$ denote channel distribution fairness, series of number of channels required, and series of number of channels received. $\beta_{P_R}$ is tolerable SINR ratio threshold value (to satisfy required quality of service level), and Prob$_{Out}$ represents the ratio of the number of outages to the number of total iterations of the greedy algorithm.

**Table 1.** Greedy Algorithm.

| | |
|---|---|
| **Input Parameters:** | • Interference graph G;<br>• Channel needs of each secondary user;<br>• Total number of channels available in the spectrum pool;<br>• Transmission power of primary and secondary users;<br>• Initial energy levels of ST users;<br>• Definitions of overlay, underlay, and guard zones;<br>• Primary and secondary overlay/underlay transmission power levels;<br>• Sensing power;<br>• Slot duration;<br>• Sensing period;<br>• Transmission period;<br>• Signal to interference plus noise ratio. |
| **Output Parameter:** | • Channel distribution fairness index. |
| **Step 1:** | • Create 4 matrices for overlay, underlay, harvesting and guard zones, respectively, each of which consists of values 1 or 0. Rows of matrices consist of STs, and columns consist of PTs (which means channels). Value of matrix element is set to 1 if ST is located in corresponding zone, otherwise 0.<br>• Create interference matrices for each ST towards each PT for overlay and underlay transmission modes, using overlay and underlay transmission powers of each ST and distance between ST and PT [44].<br>• Calculate energy levels necessary for sensing, overlay mode, and underlay mode transmissions, respectively, using Equation (2). |
| **Step 2:** | • Sort secondary users (SUs) by their degrees in the interference graph (degree = number of neighboring nodes). |
| **Step 3:** | • Select the least used channel from available channels. |
| **Step 4:** | • For the selected channel, select the node with the lowest degree from nodes not checked yet. In case two nodes having the same degree, select the one that is farther from the primary user owning that channel. |
| **Step 5:** | • Sense if the channel found in step 3 is idle or busy.<br>　**If channel==idle, then**<br>　　• check if the node found in step 4 has enough energy for overlay transmission for the channel selected in step 3.<br>　　**If yes, then**<br>　　　• calculate the total interference on the PT owning the channel if the channel is assigned.<br>　　　**If** it does not exceed the necessary interference threshold for the given SINR and neighbor nodes of the node have not already obtained this channel, **then**<br>　　　　• assign the channel to the node,<br>　　　　• update the energy level by subtracting sensing and overlay transmission energies from the initial energy,<br>　　　　• update the lists of used channels of affected nodes,<br>　　　　• if a node obtained the number of required channels, remove it from main interference graph, and proceed to step 7,<br>　　**else,**<br>　　　• check if the node lies outside the interference range of the PT owning the channel selected in step 3 and if it has enough energy for overlay transmission.<br>　　　**If yes, then**<br>　　　　• calculate the total interference on the PT owning the channel if the channel is assigned.<br>　　　　**If** the SINR condition is satisfied and the neighbor nodes of the node were not obtained by this channel already, **then**<br>　　　　　• assign the channel to the node and update the energy level by subtracting the overlay transmission energy from the initial energy.<br>　　　　**else,** if the node lies inside the interference range of the PT, check if the node has enough energy for underlay transmission.<br>　　　　　**If yes**, calculate the total interference on the channel owner PT if the channel is assigned.**If** the SINR is satisfied, **then**<br>　　　　　　• assign the channel to the node and update the energy level by subtracting sensing and underlay transmission energies from the initial energy,<br>　　　　　　• update the lists of used channels of affected nodes. If a node obtained all the required channels, remove it from main interference graph. |
| **Step 6:** | • Check if all nodes are checked towards the selected channel.<br>　**If** not, **then**<br>　　continue with step 4 by selecting another node.<br>　**else,**<br>　　continue with step 3, selecting another available channel from list. If all channels have been checked, proceed to step 7. |
| **Step 7:** | • Check if the number of nodes in the main interference graph is greater than 1.<br>　**If yes, then**<br>　　• check if no more channel can be assigned due to energy insufficiency.<br>　　**If yes, then finish.**<br>　　**else,**<br>　　　• return to step 2,<br>　**else**, **finish.** |

**Table 2.** Proposed algorithm using maximum independent sets (MIS) of interference graph.

| | |
|---|---|
| **Input Parameters:** | • Interference graph G;<br>• Channel needs of each secondary user;<br>• Total number of channels available in the spectrum pool;<br>• Transmission power of primary and secondary users;<br>• Initial energy levels of ST users;<br>• Definitions of overlay, underlay, and guard zones;<br>• Primary and secondary overlay/underlay transmission power levels;<br>• Sensing power;<br>• Slot duration;<br>• Sensing period;<br>• Transmission period;<br>• Signal to interference plus noise ratio. |
| **Output Parameter:** | • Channel distribution fairness index. |
| **Step 1:** | • Create 4 matrices for overlay, underlay, harvesting and guard zones, respectively, each of which consists of values 1 or 0. Rows of matrices consist of STs, and columns consist of PTs owning channels. Value of matrix element is set to 1 if ST is located in corresponding zone, otherwise 0.<br>• Create interference matrices for each ST towards each PT for overlay and underlay transmission modes, using overlay and underlay transmission powers of each ST and distance between ST and PT [44].<br>• Calculate energy levels necessary for sensing, overlay mode, and underlay mode transmissions, respectively, using Equation (2). |
| **Step 2:** | • Create sub graph Gi, if i = 1, Gi = G. |
| **Step 3:** | • Calculate the MIS and delete it from the subgraph. |
| **Step 4:** | • Select the least used channel from available channels and sense if the channel is idle or busy.<br>　**If channel==idle, then**<br>　　• proceed to step 5;<br>　**else,**<br>　　• proceed to step 6. |
| **Step 5:** | • Get each member node of MIS one by one and check if it has enough energy for overlay transmission for the channel checked in step 4.<br>　**If yes, then**<br>　　• calculate the total interference on the PT owning the channel if the channel is assigned.<br>　　**If** not exceeding the necessary interference threshold for a given SINR and neighbor nodes of the node have not already obtained this channel, **then**<br>　　　• assign the channel to the node,<br>　　　• update the energy level by subtracting sensing and overlay transmission energies from the initial energy,<br>　　　• update the lists of used channels of affected nodes. If a node obtained all the required channels, remove it from the main interference graph and proceed to step 7. |
| **Step 6:** | • Check if the node lies outside the interference range of the PT owning the channel selected in step 3 and if it has enough energy for overlay transmission.<br>　**If yes, then**<br>　　• calculate the total interference on the PT owning the channel if the channel is assigned.<br>　　**If** the SINR is satisfied and none of the neighbor nodes have been already obtained this channel, **then**<br>　　　• assign the channel to the node and update the energy level by subtracting sensing and overlay transmission energies from the initial energy,<br>　　　• update the lists of used channels of affected nodes. If a node obtained all the required channels, remove it from main interference graph.<br>　　**else,**<br>　　　• check if the node has enough energy for underlay transmission.<br>　　　**If yes, then**<br>　　　　• calculate the total interference on the PT owning the channel if the channel is assigned,<br>　　　　**If** the SINR is satisfied and none of the neighbor nodes have been already obtained this channel, **then**<br>　　　　　• assign the channel to the node and update the energy level by subtracting sensing and underlay transmission energies from the initial energy.<br>　　　　　• update the lists of used channels of affected nodes. If a node obtained all the required channels, remove it from main interference graph. |
| **Step 7:** | • check if all MIS nodes were checked towards the selected channel.<br>　**If yes, then**<br>　　• go to step 4 to select another available channel. If all channels have already been checked, proceed to step 8<br>　**else,**<br>　　• go to step 5. |
| **Step 8:** | • Check if the number of nodes in the main interference graph is greater than 1.<br>　**If yes, then**<br>　　• check if no more channel can be assigned due to energy insufficiency.<br>　　**If yes, then finish.**<br>　　**else,**<br>　　　• return to step 2,<br>　**else, finish.** |

**Table 3.** Proposed Algorithm Using Proposed Metric of Allocation.

| | |
|---|---|
| **Input Parameters:** | • Interference graph G;<br>• Channel needs of each secondary user;<br>• Total number of channels available in the spectrum pool;<br>• Transmission power of primary and secondary users;<br>• Initial energy levels of ST users;<br>• Definitions of overlay, underlay, and guard zones;<br>• Primary and secondary overlay/underlay transmission power levels;<br>• Sensing power;<br>• Slot duration;<br>• Sensing period;<br>• Transmission period;<br>• Signal to interference plus noise ratio. |
| **Output Parameter:** | • Channel distribution fairness index. |
| **Step 1:** | • Create 4 matrices for overlay, underlay, harvesting and guard zones, respectively, each of which consists of values 1 or 0. Rows of matrices consist of STs, and columns consist of PTs owning channels. Value of matrix element is set to 1 if ST is located in corresponding zone, otherwise 0.<br>• Create interference matrices for each ST towards each PT for overlay and underlay transmission modes, using overlay and underlay transmission powers of each ST and distance between ST and PT [44].<br>• Calculate energy levels necessary for sensing, overlay mode, and underlay mode transmissions, respectively, using Equation (2). |
| **Step 2:** | • Select the least used channel from available channels and calculate the proposed metric given in Equation (6) for each ST towards that channel, and sort the list in descending order of metric. |
| **Step 3:** | • Sense if the channel is idle or busy.<br>   **If channel==idle, then**<br>      • proceed to step 5;<br>   **else,**<br>      • proceed to step 6. |
| **Step 4:** | • Get the node with highest metric of the list -which is not checked yet- and check if it has enough energy for overlay transmission for the channel checked in step 3.<br>   **If yes, then**<br>      • calculate the total interference on the PT owning the channel if the channel is assigned.<br>     **If** not exceeding the necessary interference threshold for a given SINR and neighbor nodes of the node have not already obtained this channel, **then**<br>        • assign the channel to the node,<br>        • update the energy level by subtracting sensing and overlay transmission energies from the initial energy,<br>        • update the lists of used channels of affected nodes. If a node obtained all the required channels, remove it from main interference graph and proceed to step 6. |
| **Step 5:** | • Get the node with highest metric of the list -which is not checked yet- and check if it lies outside the interference range of the PT owning the channel selected in step 3.<br>   **If yes, then**<br>      • check if the node has enough energy for overlay transmission.<br>     **If yes, then**<br>        • calculate the total interference on the PT owning the channel if the channel is assigned.<br>       **If** not exceeding the necessary interference threshold for a given SINR and neighbor nodes of the node have not already obtained this channel, **then**<br>          • assign the channel to the node,<br>          • update the energy level by subtracting sensing and overlay transmission energies from the initial energy.<br>          • update the lists of used channels of affected nodes. If a node obtained all the required channels, remove it from main interference graph.<br>     **else,**<br>        • check if the node has enough energy for underlay transmission.<br>       **If yes, then**<br>          • calculate the total interference on the PT owning the channel if the channel is assigned.<br>        **If** not exceeding the necessary interference threshold for a given SINR and neighbor nodes of the node have not already obtained this channel, **then**<br>            • assign the channel to the node,<br>            • update the energy level by subtracting sensing and underlay transmission energies from the initial energy.<br>            • update the lists of used channels of affected nodes. If a node obtained all the required channels, remove it from main interference graph. |
| **Step 6:** | • Check if all STs are tested towards the selected channel.<br>   **If yes, then**<br>      • continue with step 3 by selecting another available channel. If all channels have already been checked, proceed to step 7.<br>     • **else,**<br>        • return to step 4. |
| **Step 7:** | • Check if the number of nodes in the main interference graph is greater than 1.<br>   **If yes, then**<br>      • check if no more channel can be assigned due to energy insufficiency.<br>     **If yes, then finish.**<br>     **else,**<br>        • return to step 2 and restart channel allocation with new updated energy levels.<br>   **else**, **finish.** |

**Table 4.** Simulation parameters.

| Symbol | Description | Value |
|--------|-------------|-------|
| T | Slot Duration | 1 ms |
| $\tau$ | Sensing Duration | 0.002 ms |
| T - $\tau$ | Transmission Duration | 0.098 ms |
| E | Initial Energy | random value in range [0, max(E)] |
| $P_S$ | Sensing Power | 110 mW |
| $P_O$ | Overlay Transmit Power | 50 mW |
| $P_U$ | Underlay Transmit Power | 30 mW |
| SINR | Signal to Interference Plus Noise Ratio | dB |
| $E_H$ | Harvested Energy | mJ |
| $r_h$ | Radius of Harvesting Zone | 33 m |
| $r_g$ | Radius of Guard Zone | 75 m |
| $N_o$ | Guassion Noise with Zero Mean | $10^{-7}$ mW |
| $P_T$ | Primary User' s Power | 1 W |
| $e_t$ | Residual Energy at the Beginning of Time Slot t | mJ |
| $e_o$ | Energy Necessary to Make Overlay Transmission | 5.12 mJ |
| $e_u$ | Energy Necessary to Make Underlay Transmission | 3.16 mJ |
| $\eta$ | Harvesting Conversion Efficiency | 0.75 |
| n | Max Number of Available Channels for each ST | 3 |
| $\alpha$ | Path-loss Exponent | 0.75 |

However, because both greedy and proposed methods check interference condition and allocate channels so that interference does not exceed the threshold, $ProbOut(\beta_{P_R})$ will always be 0 for proposed methods. The ratio of STs in the HZ to the total number of STs is calculated to obtain information about the percentage of STs that can harvest energy.

$$Rat_{st} = N_{st}/T_{st}, \tag{10}$$

where $Rat_{st}$, $N_{st}$, and $T_{st}$ show the ratio of STs that can do harvest, number of STs in harvesting zone, and total number STs, respectively. In order to show fairness of proposed schemes, 10 different experiments were run, and all experimental results were averaged over 1000 independent runs. The experimental results are as follows:

**Experiment 1.** *In the first experiment, the channel distribution performance (ChDistFairness) obtained according to Equation (9) for various SINR (outage) threshold values is examined. As mentioned before, since allocations are applied by checking and controlling interference levels each time, $ProbOut(\beta_{P_R})$ will always be 0.*
*Parameters used in this experiment can be summarized as:*

- *Number of PTs is 2, which means the number of available channels is 2 and the number of idle channels varies randomly between 0 and 2.*
- *The Poisson distribution density of STs was about 0.0015 in the $100 \times 100~m^2$ area around the PT.*
- *Average estimated percentage of STs in HZ is about 21%, according to Equation (10).*

**Results can be commented on as below:**

- Figure 4 shows that with increasing SINR (outage) threshold values, the channel distribution fairness factor calculated using Equation (9) increases from 0.0387 to about 0.039, which is close to the upper limit of ChDistFairness of 0.05. This proves that as the number of STs inside the 100 $\times$ 100 m$^2$ area increases, the co-channel interference between neighboring nodes increases and, additionally, more STs exist in the guard zone of a PT, which means they could not make use of those channels. As such, ChDistFairness is not much affected by SINR changes.
- As the average percentage of STs in HZ was 21%, which implies almost every 1 of 5 STs can harvest energy, ChDistFairness improves, especially in the first three time slots since some STs harvest energy and use it for their sensing and transmission.

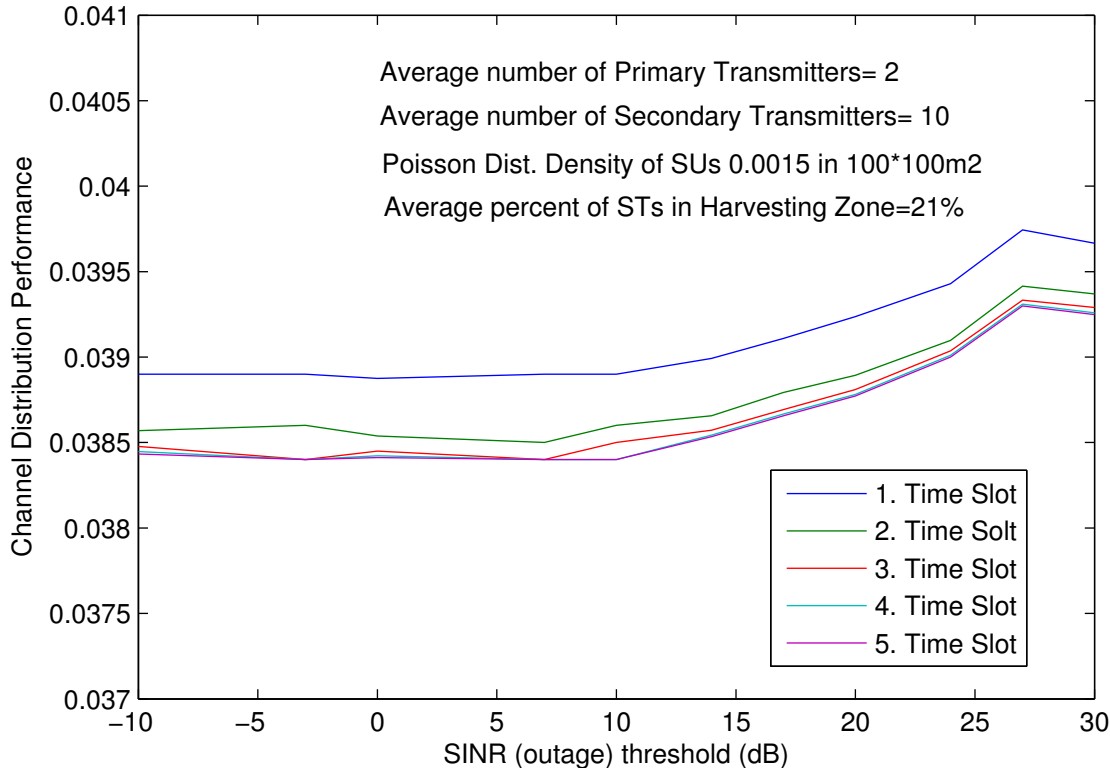

**Figure 4.** ChDistFairness against different SINR (threshold) values for Poisson distribution density of STs of 0.0015 on the area of $100 \times 100$ m² around the PT and average percentage of STs in harvesting zone of about 21%.

**Experiment 2.** *In the second experiment, again, ChDistFairness for various SINR (outage) threshold values is examined.*

*Parameters used in this experiment can be summarized as:*

- *The total number of PTs and STs are increased to 4 and 16, respectively. That means number of available channels is 4 and number of idle channels varies randomly between 0 and 4.*
- *However, compared to Experiment 1, the density of STs in a $200 \times 200$ m² area distributed around the PT decreases to about 0.0005, and the percentage of harvestable STs decreases to about 4%.*

   **Results can be on commented as below:**

- As depicted in Figure 5, with increasing SINR (outage) threshold values, ChDistFairness increases from 0.015 to about 0.021, which is far below the upper limit of 0.05, which is quite good.
- Although the total number of STs increases, the number of STs inside the $100 \times 100$ m² area falls, thereby decreasing in the guard zone, as well. Since the distance between STs increases (because of density decrease), the average number of neighbors of STs decreases, thereby reducing the co-channel interference between neighboring nodes, too, which allows more channels to be shared.
- When the SINR (outage) threshold increases, the interference constraint on PTs becomes more dominant, so ChDistFairness drops accordingly.
- The average percentage of STs in the HZ is about 4%; and, due to those harvesting nodes, ChDistFairness still improves, especially in the first, second, and third time slots due to utilizing the harvested energy.

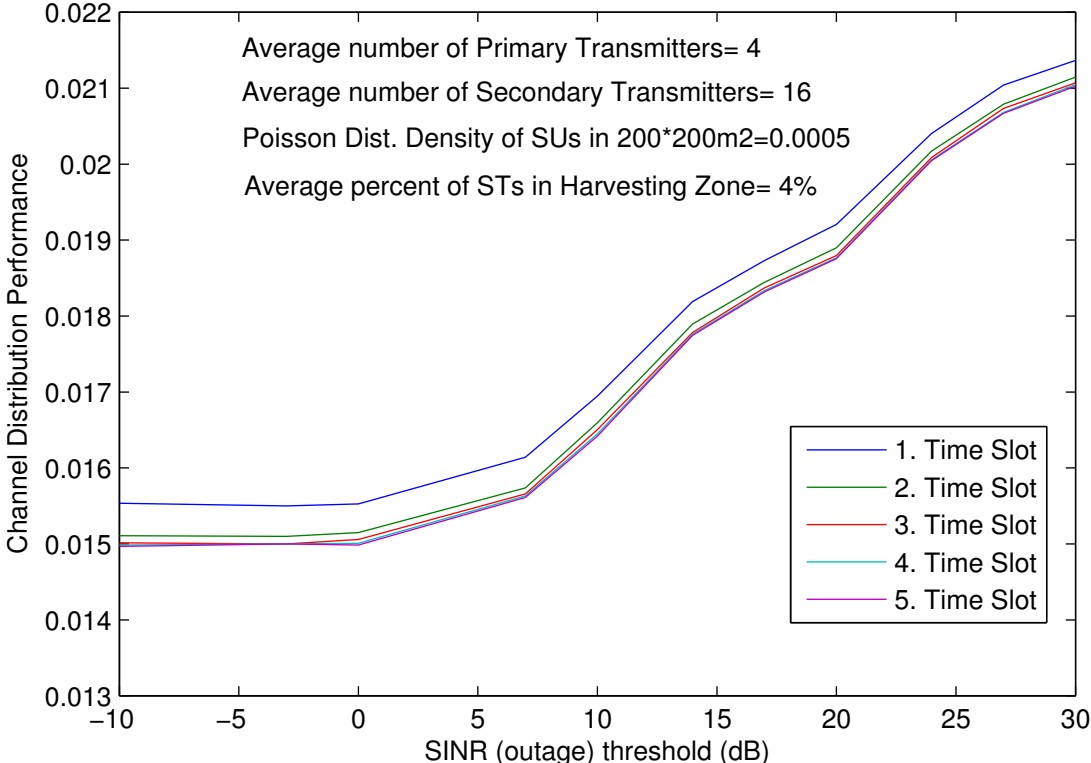

**Figure 5.** ChDistFairness against different SINR (threshold) values for Poisson distribution density of STs of 0.0005 in a $200 \times 200$ m$^2$ area around the PT and average percentage of STs in harvesting zone of about 4%.

**Experiment 3.** *ChDistFairness for various SINR (outage) threshold values is examined.*
*Parameters used in this experiment can be summarized as:*

- *The total number of PTs and STs are increased again to 6 and 26, respectively.*
- *The density of the STs distributed around the PT decreases to about 0.0005 in $300 \times 300$ m$^2$ yet, which means the percentage of harvesting STs falls to around 3%.*

**Results can be commented on as below:**

- As shown in Figure 6, with increasing SINR (outage) threshold values, ChDistFairness values vary from 0.0065 to about 0.015, which seems to be better than the previous results. This is because number of available channels (PTs) increases from 4 to 6.
- As the total number of STs increases, the number of STs inside the $100 \times 100$ m$^2$ area decreases, thereby decreasing in the guard zone, as well, which results in an increase in the probability of channels usage.
- As STs move away from each another due to the density decrease, the co-channel interference drops, which means more channels can be shared. Again, as expected, channel distribution performance worsens (so ChDistFairness raises) when the SINR (outage) threshold increases, accordingly, due to the interference constraint.
- In accordance with the ST harvesting capability in HZ, which is about 3%, ChDistFairness drops again with increasing time slots.

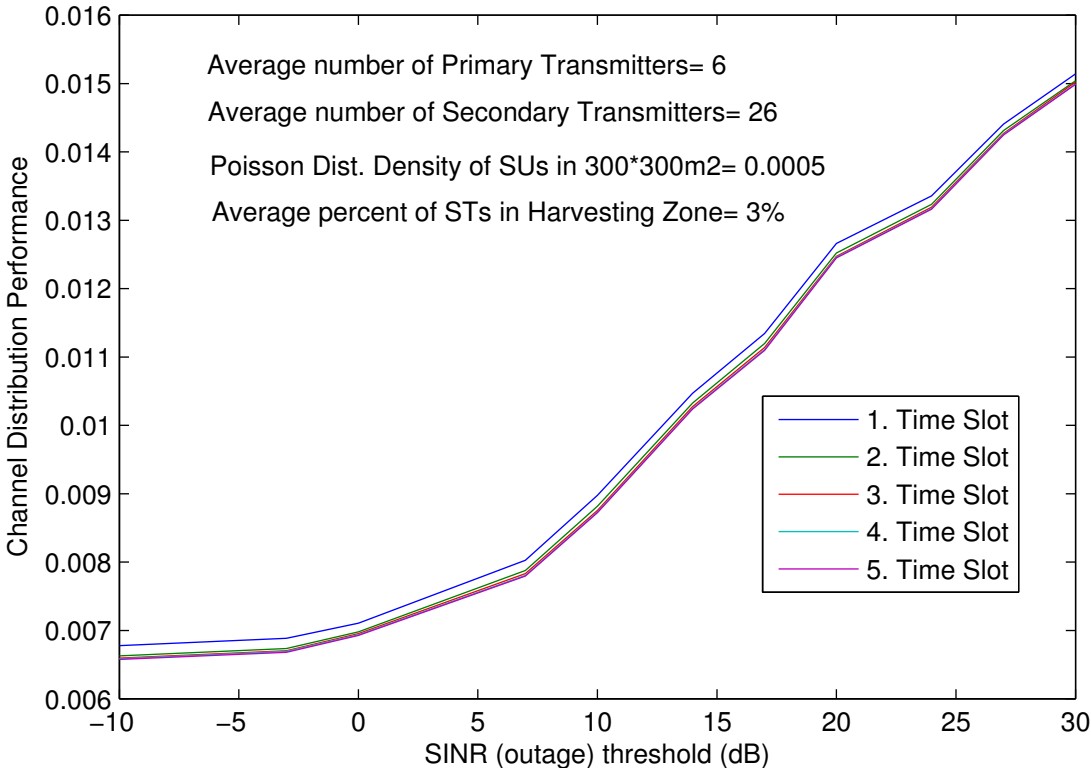

**Figure 6.** ChDistFairness against different SINR (threshold) values for Poisson distribution density of STs of 0.0005 in an area of $300 \times 300$ m$^2$ around the PT and average percentage of STs in harvesting zone of about 3%.

**Experiment 4.** *ChDistFairness for various SINR (outage) threshold values is examined. Parameters used in this experiment can be summarized as:*

- *The density of the STs distributed around the PT is decreased to about 0.0002 in $400 \times 400$ m$^2$.*
- *The total number of PTs and STs are 6 and 25.*

**Results can be commented on as below:**

- Figure 7 shows again that with increasing SINR (outage) threshold values, the ChDistFairness values vary between 0.0065 and about 0.016.
- Similar to previous results, due to density decrease, the number of STs inside the guard zone decreases. That is why the co-channel interference drops and results in more channels to be allocated.
- Again, as expected, ChDistFairness increases when the SINR (outage) falls. In accordance with the harvesting capability of STs in the HZ, which is about 3%, ChDistFairness falls again with progressive time slots.

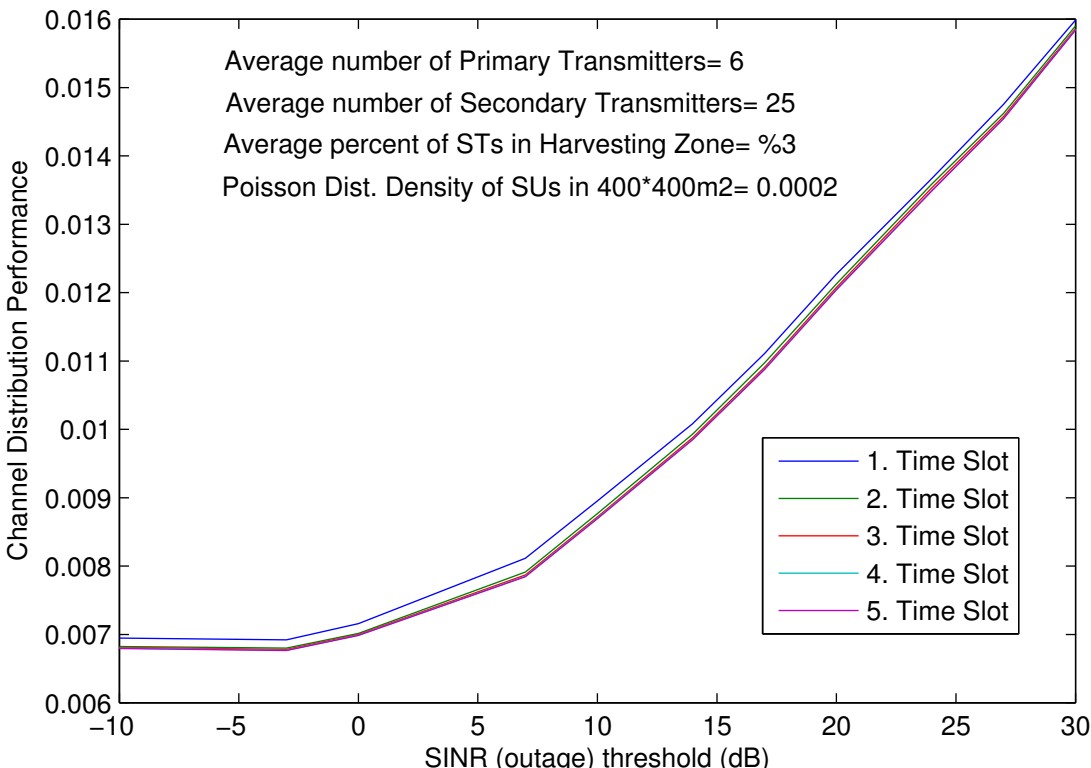

**Figure 7.** ChDistFairness against different SINR (threshold) values for Poisson distribution density of STs of 0.0002 on the area of $400 \times 400$ m$^2$ around the PT and average percentage of STs in the harvesting zone of about 3%.

**Experiment 5.** *ChDistFairness for various SINR (outage) threshold values is examined.*
*Parameters used in this experiment can be summarized as:*

- *The numbers of PTs and STs are further increased to 8 and 32.*
- *The Poisson distribution density of STs is about 0.0003 in the $400 \times 400$ m$^2$ area around the PT.*
- *Average estimated percentage of STs in HZ is about 2%.*

**Results can be commented on as below:**

- Comparison of the results in Figure 8 with the previous experiment shows that ChDistFairness varies between 0.005 and 0.013, which indicates better performance. Even though the average percentage of STs in HZ is a little lower and density is slightly higher, more channels seem to be allocated since more channels are available to be shared due to the increase in the number of PTs from 6 to 8.
- On the other hand, it is observed that ChDistFairness does not much change against increasing time slots. This is because number of total STs increases and, even if harvesting is applied, this cannot improve ChDistFairness much since number of available channels per ST is more restricted.
- In addition, average percentage of STs in the harvesting zone decreases from 3% to 2%, which reduces energy harvesting capability that results in decreasing on improvement of ChDistFairness.

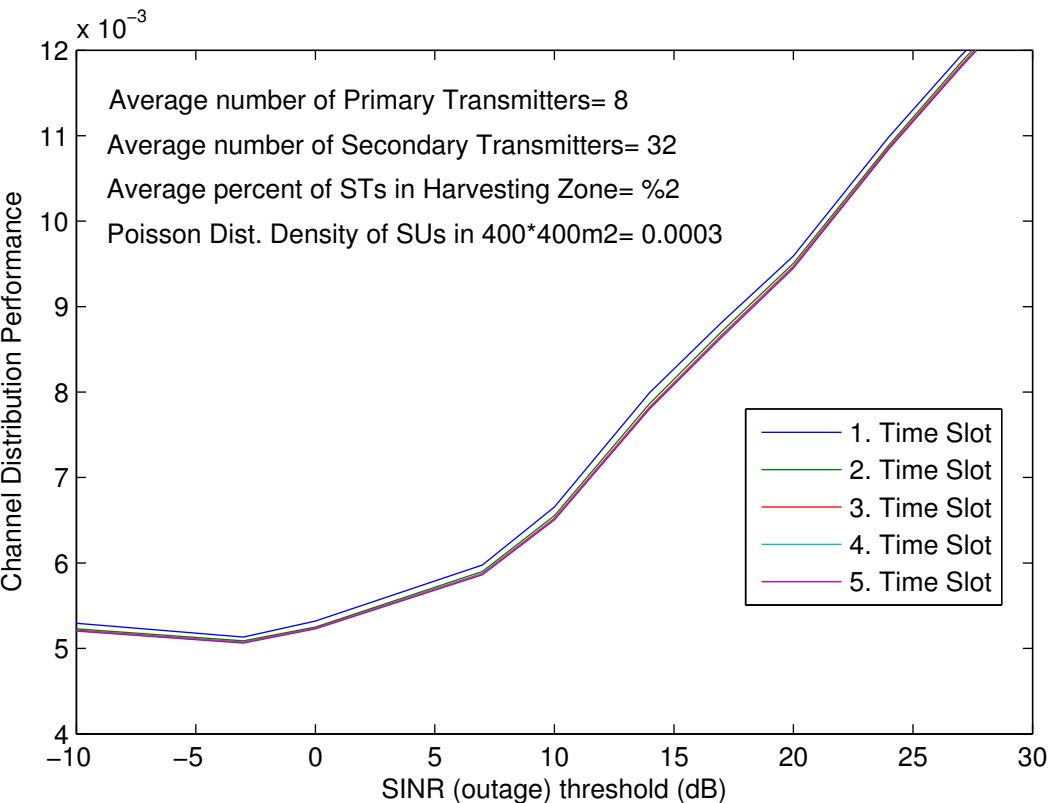

**Figure 8.** ChDistFairness against different SINR (threshold) values for Poisson distribution density of STs of 0.0003 in a $400 \times 400$ m$^2$ area around the PT and average percentage of STs in the harvesting zone of about 2%.

**Experiment 6.** *In this experiment, ChDistFairness is compared against increasing time slots.*
*Parameters used in this experiment can be summarized as:*

- *Average percentages of STs in the HZ is 21%, and the number of PTs and STs are 2 and 20, respectively, when the SINR (outage) threshold fixed to 1.*
- *Average percentages of STs in the HZ is 4%, and the number of PTs and STs are 4 and 16, respectively, when the SINR (outage) threshold fixed to 1.*
- *Average percentages of STs in the HZ is 3%, and the number of PTs and STs are 6 and 25, respectively, when the SINR (outage) threshold fixed to 1.*
- *Average percentages of STs in the HZ is 2%, and the number of PTs and STs are 8 and 30, respectively, when the SINR (outage) threshold fixed to 1.*

**Results can be commented on as below:**

- The results are depicted in Figure 9. For each of the cases, ChDistFairness decreases (improves) in the first 3–4 time slots as a result of harvesting capability. This effect happens most in the case with the highest percentage of STs in HZ of 21% (blue curve) where harvesting is more utilized than the others (next experiment depicts this effect more clearly).
- The channel distribution performance is best when the number of PTs is highest (PTs = 8, red curve) as the highest number of channels are available for STs.

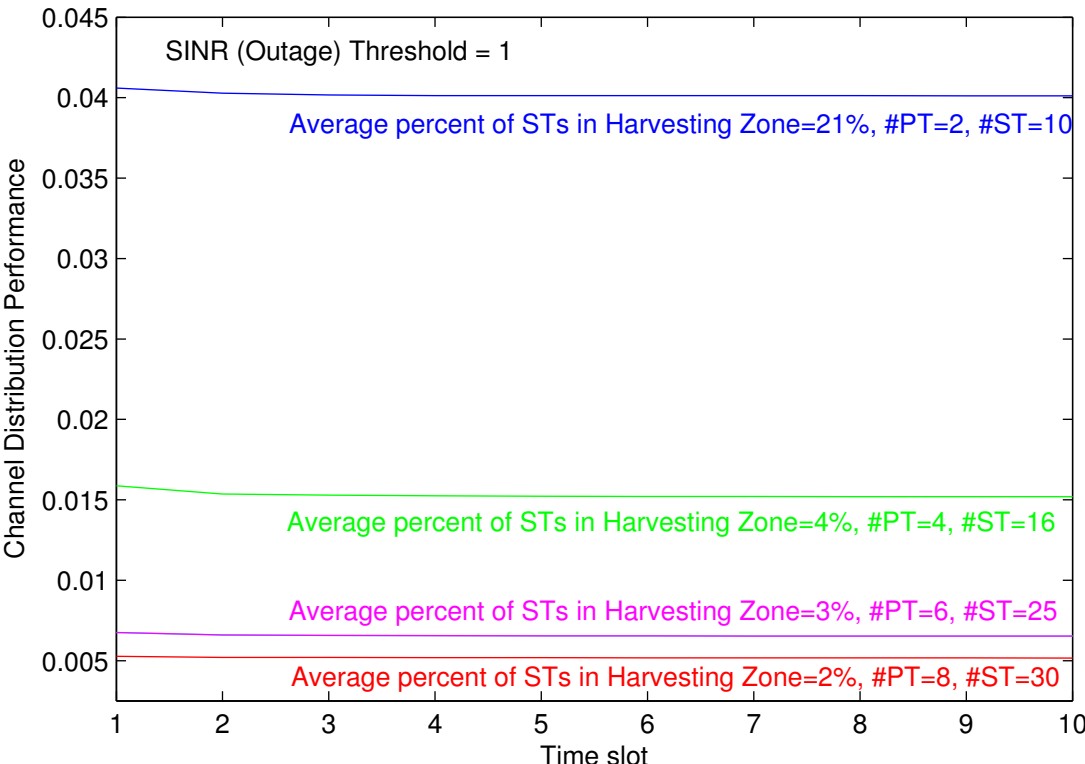

**Figure 9.** ChDistFairness against time slots for different percentages of STs in the harvesting zone for constant SINR (Outage) threshold of 1.

**Experiment 7.** *In the seventh experiment, ChDistFairness is compared against increasing time slots. Parameters used in this experiment can be summarized as:*

● *Average percentage of STs in HZ is fixed to 21% and number of PTs and STs are 2 and 10, respectively, when the SINR (outage) threshold fixed to 1.*

**Results can be commented on as below:**

● As shown in Figure 10, ChDistFairness decreases until the fifth time slot via harvesting capability when the SINR (outage) threshold is fixed to 1.

**Experiment 8.** *In the eighth experiment, ChDistFairness levels are examined for different initial energy levels. Parameters used in this experiment can be summarized as below:*

● *Three different experiments run. In the first experiment, at each run, initial energy levels are randomly set between (0, 5 mj); in the second experiment, they are set between (0, 10 mj), and at third experiment, they are set between (0, 15 mj) sequentially, which means possible maximum values are 5 mj, 10 mj, and 15 mJ.*
● *The SINR (outage) threshold is fixed to 10.*
● *The average percentage of STs in the HZ is fixed to 4%.*

**Results can be commented as below:**

● As expected, for the case with lowest initial energy level between (0, 5 mj), STs are able to allocate fewer channels since their energy runs short of fastest. The results are shown in Figure 11.
● With increasing initial energy level, STs have more ability to utilize channels against time slots since energy is consumed slower. So, results given in blue seem better than red.

- In the third case, when initial energy level is set between (0, 15 mj), ChDistFairness results become the best since STs are most capable to utilize channels because of having more energy.

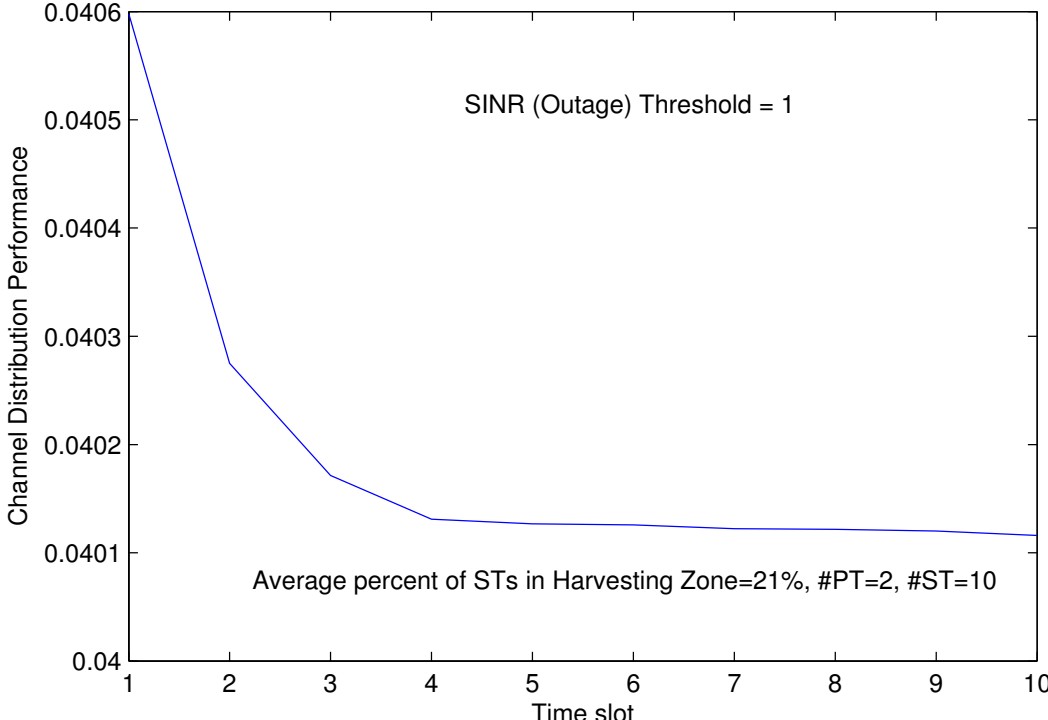

**Figure 10.** ChDistFairness against time slots for constant percentage of STs in the harvesting zone of 21% and constant SINR (Outage) threshold of 1.

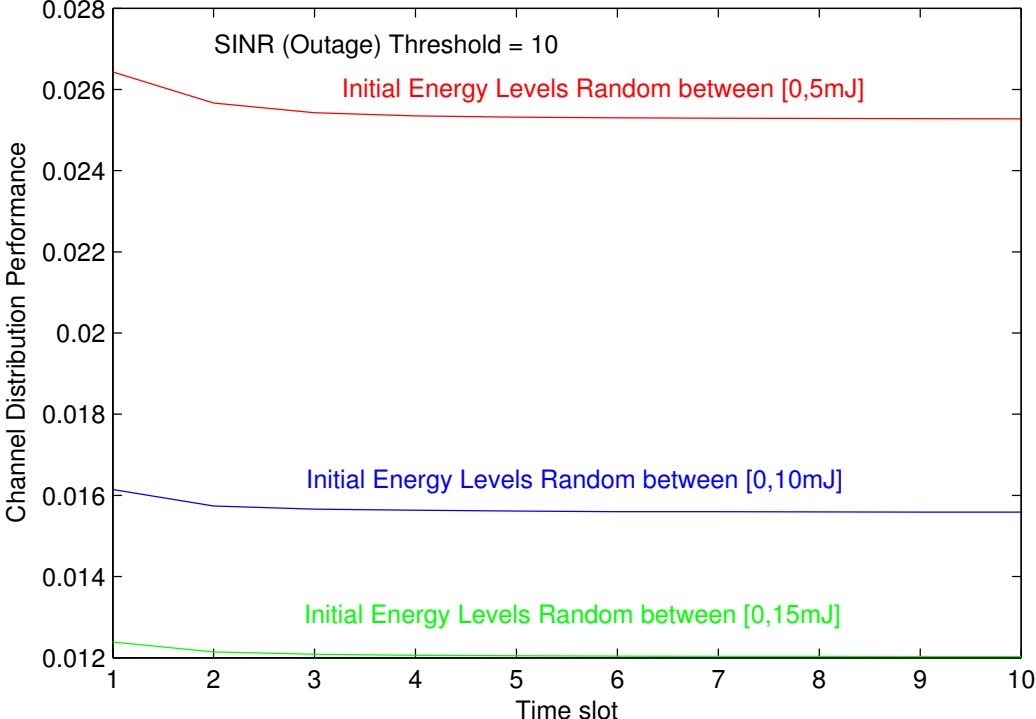

**Figure 11.** ChDistFairness against time slots for different initial energy levels of STs.

**Experiment 9.** *In the ninth experiment, ChDistFairness is again compared against increasing time slots, however, this time, for varying PT transmission power levels.*
*Parameters used in this experiment can be summarized as:*

- *PT transmission power level is set to 1000 mW when the SINR (outage) threshold fixed to 1.*
- *PT transmission power level is set to 500 mW when the SINR (outage) threshold fixed to 1.*

**Results can be commented on as below:**

- Results are depicted in Figure 12. When PT power is higher, which means the numerator in Equation (7) is higher, the sum of interferences in the denominator is allowed to be higher when keeping the same SINR (outage) threshold value. Therefore, STs being allowed to create more interference on primary receiver have more opportunity to utilize channels, which results in higher channel distribution performance (lower ChDistFairness).

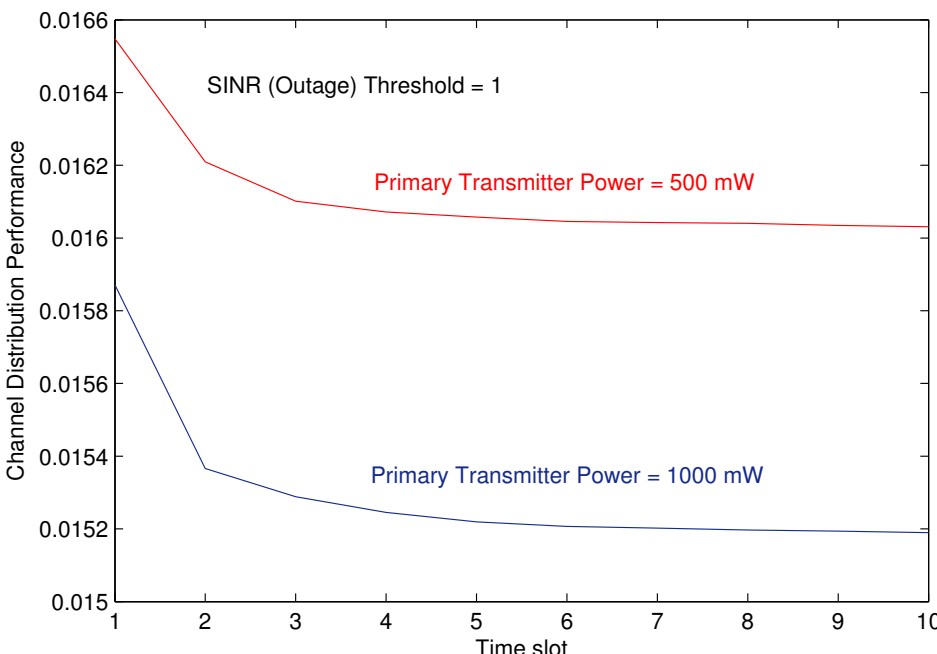

**Figure 12.** ChDistFairness against time slots for PT transmission power levels of 500 and 1000 mW.

**Experiment 10.** *In the last experiment, ChDistFairness results are compared for greedy, proposed maximum independent set (MIS)-based, and proposed metric based allocation mechanisms against time slots under the same average percentage of STs in HZ. Parameters used in this experiment can be summarized as below:*

- *PT transmission power, ST overlay, and underlay mode transmission powers are set to 1000 mW, 50mW, and 30 mW, respectively, when the SINR (outage) threshold fixed to 1 and number of PTs and STs are 4 and 16, respectively.*
- *PT transmission power, ST overlay, and underlay mode transmission powers are set to 1000 mW, 50 mW, and 30 mW, respectively, when the SINR (outage) threshold fixed to 100 and number of PTs and STs are 4 and 16, respectively.*
- *PT transmission power, ST overlay, and underlay mode transmission powers are set to 1000 mW, 50 mW, and 30 mW, respectively, when the SINR (outage) threshold fixed to 1000 and number of PTs and STs are 4 and 16, respectively.*

**Results can be commented on as below:**

- Figure 13–15 depict the results for differing SINR (outage) threshold values. As shown, for each of these cases, the performance of the greedy algorithm (blue curve) is the worst since it does not take advantage of energy harvesting mechanism nor make prioritization-based allocation.
- The proposed algorithm using MIS-based allocation (green curve) performs better allocation than the greedy method since STs store energy via harvesting.
- Finally, as the red curve depicts, the proposed algorithm using the proposed metric given by Equation (6) performs the best (the lowest ChDistFairness values between the 3 algorithms). This is because this algorithm allocates channels taking into account prioritization of those nodes having more initial energy, more harvesting capability, and also multi-channel usage ability, thus owning higher metric value, which ensures more channels be utilized against increasing time slots.

Consequently, all these simulation results prove, of the 3 algorithms, the proposed scheme using the proposed metric performs allocation with the smallest ChDistFairness values, which explains how it maximizes allocation performance. This verifies the superiority of the proposed mechanism.

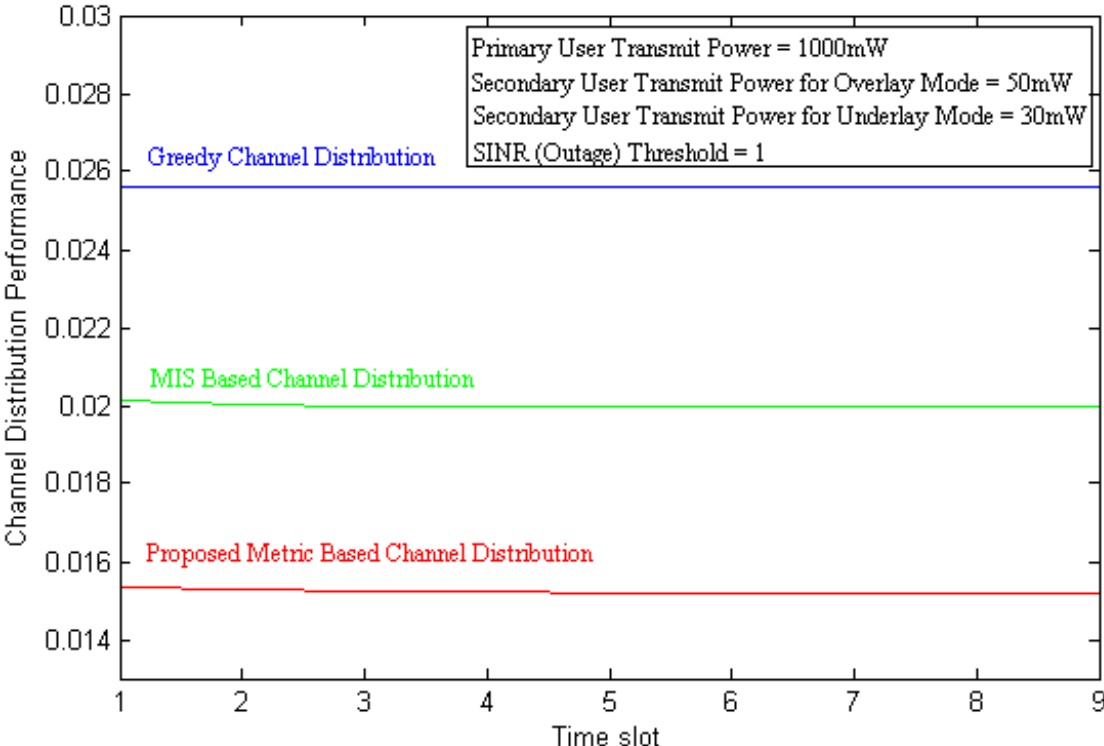

**Figure 13.** ChDistFairness against time slots for 3 algorithms when the SINR (outage) threshold is 1.

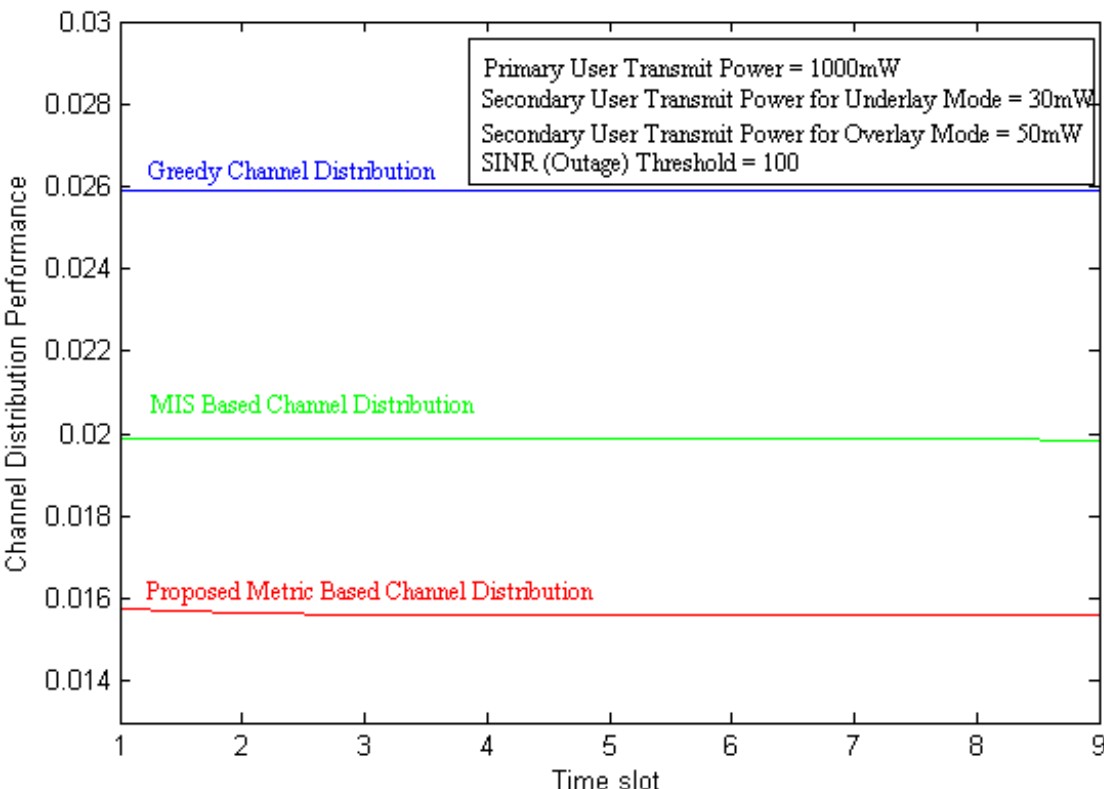

**Figure 14.** ChDistFairness against time slots for 3 algorithms when the SINR (outage) threshold is 100.

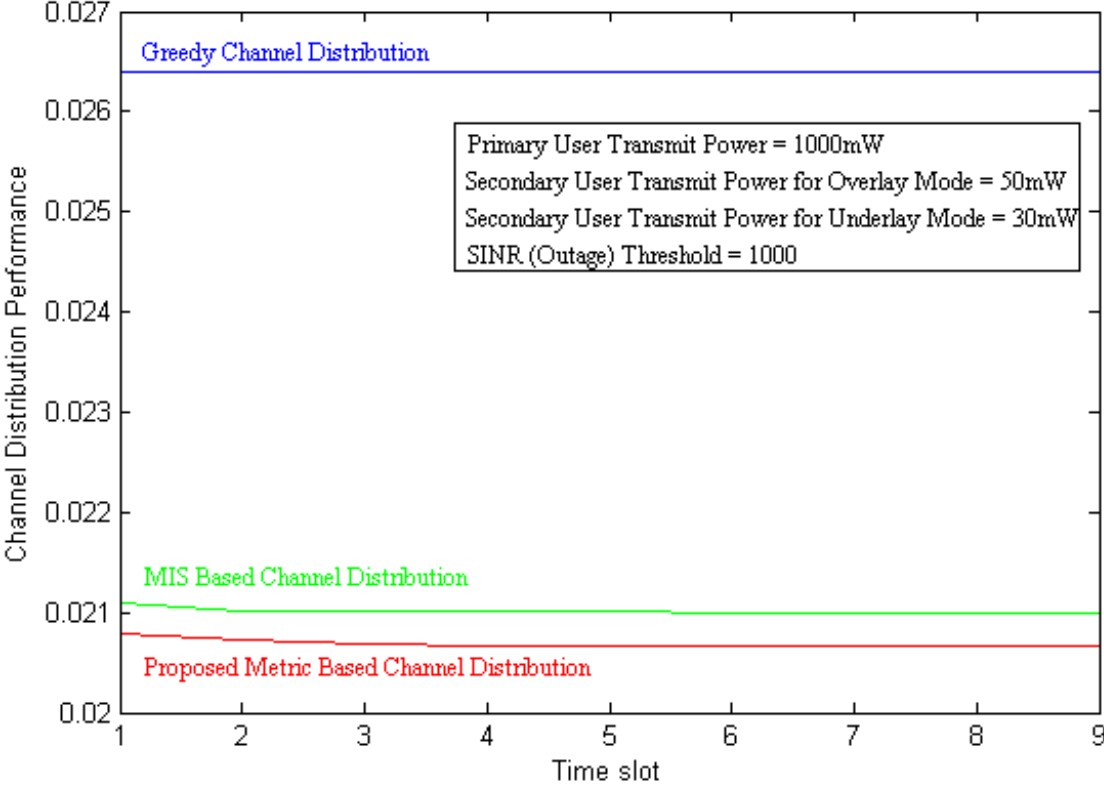

**Figure 15.** ChDistFairness aginst time slots for 3 algorithms when the SINR (outage) threshold is 1000.

## 4. Conclusions

This study focused on the performance maximization of an m-channel distribution of secondary users with harvesting capability in a hybrid cognitive radio network under the SINR constraint on primary receivers. A novel algorithm was proposed to allocate multiple channels in a hybrid cognitive radio network where STs can harvest from RF signals. To maximize allocation performance, a metric was proposed for each ST, which enables their prioritization during allocation of channels. To the best of our knowledge, this is the first work to investigate the assignment of multi-channels to SUs with multi-channel transmission capability and harvesting ability in a hybrid cognitive radio network to maximize channel allocation performance. For this purpose, the circular transmission area of an ST was divided into four zones: overlay and underlay mode areas, as well as guard and harvesting zones. Accordingly, the STs had two transmission modes based on their distance to channel owner PTs: underlay and overlay modes. The STs located inside HZs can harvest using power conversion circuits that draw DC power from the received RF signals. Each ST was assumed to be equipped with the same number of harvesters and wireless interfaces as the required number of channels.

In the simulations, a greedy hybrid allocation algorithm, a proposed MIS-based hybrid allocation mechanism with harvesting capability, and the proposed metric-based hybrid allocation scheme with harvesting capability were investigated, and the results were examined and compared. The simulation results indicate that addition of harvesting capability to allocation scheme markedly improves m-channel allocation performance when comparing MIS-based and the proposed metric-based allocation mechanisms with the greedy one. The allocation results based on sorting nodes according to the proposed metric are better than results obtained with MIS-based allocation, even though both make use of energy harvesting. The results prove the superiority of the proposed mechanism with the proposed metric because, if an ST hosted

- with more initial energy,
- having energy harvesting capability, and
- equipped with multi-channel transmission ability

is prioritized for channel assignment, metric ChDistFairness will be minimized, thereby maximizing allocation performance. Another important consequence is that outputs regarding channel distribution performance against number of nodes in HZ, which is the density of the Poisson distributed nodes, show that, as the number of nodes in HZ increases, ChDistFairness decreases faster than the case with fewer nodes in HZ, especially in the first four to five time slots for a constant interference constraint (SINR threshold value), which depicts the importance of harvesting once more.

**Funding:** This research received no external funding.

**Acknowledgments:** I would like to thank the anonymous reviewers for their insightful comments.

**Conflicts of Interest:** The authors declare no conflict of interest.

## Abbreviations

The following abbreviations are used in this manuscript:

| | |
|---|---|
| 3GPP | 3rd Generation Partnership Project |
| $\beta_{P_R}$ | Tolerable SINR ratio threshold at primary receiver $P_R$ |
| ChDistFairness | Channel Distribution Fairness |
| CN | ChannelNeed |
| CR | Cognitive Radio |
| CRN | Cognitive Radio Networks |
| CWPCN | Cognitive wireless powered communication network |
| $d_{(P_T,P_R)}$ | Distance between primary transmitter $P_T$ and primary receiver $P_R$ |
| $d_{(C_i,P_R)}$ | Distance between secondery user $s_i$ and primary receiver $P_R$ |
| DOAJ | Directory of open access journals |

| EL | Energy Level |
| EH | Energy Harvesting |
| HPPP | Homogeneous Poisson Point Process (HPPP) |
| HZ | Harvesting Zone |
| MAC | Medium Access Control |
| MDP | Markov Decision Process |
| MDPI | Multidisciplinary Digital Publishing Institute |
| MIS | Maximum Independent Set |
| $N_{st}$ | Toal number of STs in Harvesting Zone |
| LD | linear dichroism |
| LTE | Long-term evolution technology |
| PM | PriorityMeasure |
| PR | Primary Receiver |
| $\text{Prob}_{Out}$ | Outage Probability |
| PT | Primary Transmitter |
| PU | Primary User |
| $P_{U_{th}}$ | Tolerable interference threshold at the primary receiver for $\beta_{P_R}$ |
| $\text{Rat}_{st}$ | Ratio of STs in Harvesting Zone |
| RF | Radio-frequency |
| SINR | Signal to Interference plus Noise Ratio |
| ST | Secondary Transmitter |
| SU | Secondary User |
| $T_{st}$ | Total number of STs |
| TLA | Three letter acronym |

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
