# Peer review of "Throughput Optimization of Multichannel Allocation Mechanism under Interference Constraint for Hybrid Overlay/underlay Cognitive Radio Networks with Energy Harvesting"

_electronics, doi:10.3390/electronics9020330_

Round 1
Reviewer 1 Report
The work presented in the submitted manuscript is very interesting. The author provided an over-detailed introduction. I have two major concerns and some minor comments
Major comment:
A-the definition of overlay CR in page 4, is not right, the provided definition is for interweave mode please to the below references
1- Awin, Faroq A., et al. "Technical issues on cognitive radio-based Internet of Things systems: A survey." IEEE Access 7 (2019): 97887-97908.
2- Awin, Faroq, et al. "Blind spectrum sensing approaches for interweaved cognitive radio system: A tutorial and short course." IEEE Communications Surveys & Tutorials 21.1 (2018): 238-259.
The author is encouraged to add them to the list of references.
B-The author briefly discussed his simulation results, more elaborations are required. So, the reader can understand your work.
Minor comments
1-Figure 1 has been mentioned in Page 3 and displayed on page 6, it should be displayed in Page 3 or 4.
2- The author used first-person writing, such as "I focused" on Page 3, and "I examined" on page 18. It is better to use third-person like " this study focuses on"
3- It recommended to "author et al [17]" by the study in [17]. This has been repeated many times on Page 2.
4- Please enlarge the size of figure 1,2,3.
5- it is good to display your algorithm as a one-column table.
6- The author is encouraged to remove the big spaces in Pages 12,13, 14
7- The sizes of the figures should be consistent.
8- Abbreviations list is needed in such articles.
9-The table of simulation parameters should be shown in the section of simulation results.
Author Response
Dear Reviewer,
Here are my replies:
The proposed allocation schemes are reorganized in tabular form so that they can be more readable and understandable. Also, the font size in the tables made a little smaller such that each algorithm fits to one page.
Note: Corrections and updates in the journal are highlighted by yellow too ("Please see the attachment as well").
MAJOR Comments:
A- The definition of overlay CR is corrected as defined in the below references and definitions are referenced by adding them to the list of references.
1- Awin, Faroq A., et al. "Technical issues on cognitive radio-based Internet of Things systems: A survey." IEEE Access 7 (2019): 97887-97908.
2- Awin, Faroq, et al. "Blind spectrum sensing approaches for interweaved cognitive radio system: A tutorial and short course." IEEE Communications Surveys & Tutorials 21.1 (2018): 238-259.
B- Simulation results are explained in more detail. Each experiment' s purpose, parameters used and comments are described separately and clearly in detail.
MINOR Comments:
Figure 1 has been placed on page 4.
All declarations like "I focused.." have been modified as "this study", "this journal" etc..
Declarations like "author et al" have been corrected.
Sizes of figure 1,2 and 3 have been enlarged.
All algorithms have been organized in tabular one column form.
Big spaces have been removed.
All figure sizes have been consistent.
List of abbreviations which is at the end of the journal is updated.
The table of simulation parameters has been placed in results and discussion section.
Thank you for your invaluable comments. Best Regards.

Reviewer 2 Report
In this paper, the author considered a cognitive radio network in which secondary users harvest energy from primary users. Currently, both of cognitive radio and energy harvesting techniques are popular topics so that some readers may be interested in this paper. However, I think that a major revision of this paper has to be made by the reasons as follows:
1. The author proposed two schemes for allocating channels to secondary users. However, the schemes are so "discursively" stated that it is too hard to understand them.
2. This paper contains some mathematical expressions. However, they are not properly written. Some variables are never defined. For example, (9) contains a number of undefined variables. What are Ch_{need}, Ch_{obt}, S, and SINR_{Thr}? Moreover, some variables seem to be doubly defined. For example, E is used to represent energy and edge as well. Variables should be carefully printed. d_{(P_{T},P_{R})} is definitely not equal to d(P_{T},P_{R}). Upright and italic characters should be distinguished. Such an ambiguity in mathematical expressions must be cleared for the readers to understand the proposed schemes.
3. The proposed schemes assume that every secondary user exactly know about everything of the network; locations of primary users and other secondary users, available channels, energy things, and so on. How can a secondary user get the information about the network? Is it practically possible?
4. The proposed schemes concern the distance between primary transmitter and secondary transmitter. Such a distance evidently affects the amount of harvested energy. Regarding interference, however, the distance between primary receiver and secondary transmitter as well as the distance between primary transmitter and secondary receiver should be considered. How are such distances used in the paper?
5. The abstract states that schemes of maximizing allocation performance are proposed. In all the figures, however, simulation results only demonstrate that the proposed schemes are relatively better in "fairness". What is the aim of the proposed schemes?
6. (9) is used to measure "fairness" in this paper. However, the reason why (9) is used is not properly stated.
7. (6) is an important factor in the proposed schemes. However, the meaning of (6) is not clearly addressed. Moreover, (6) contains some constants, say, 0.4 and 0.3. How did you get the numbers?
Author Response
Dear Reviewer,
Here are my replies:
Note: Corrections and updates in the journal are highlighted by yellow too ("Please see the attachment as well").
The proposed allocation schemes are reorganized in tabular form so that they can be more readable and understandable. Also, the font size in the tables made a little smaller such that each algorithm fits to one page. Variables are revised and undefined variables are explained in detail. Symbol of energy level is changed with symbol EL. Faulty mathematical expressions are corrected.
3. In order to determine in which area an ST locates, the mobility model of
an ST is not specified as the statistical information of ST locations is only needed in my analysis. Random walk mobility model [35] may be used to obtain statistical information of locations. Based on the statistical information of ST locations, the probability that an ST exists in any of areas can be determined. However, it is not the main focus of this paper which approach to use to obtain these probabilities. Therefore, these probabilities are obtained by open approaches based on the mobility model, the area in each zone, and the initial probability distribution of ST. ST knows the area where it locates by the received power of RF signals and the interference threshold of primary users. On the other hand, this journal supposes there exists a centralized server in the CR network and SUs broadcast their locations, available channels and energy levels to that server which is called spectrum server as well. Therefore, flow routing, and spectrum/energy management are straightforward and organized. During flow of algorithm, SUs need to communicate with spectrum server, and the center should inform about the energy level/allocation results to users. It is supposed the interaction between SUs and spectrum server does not have any impact on spectral efficiency (communication occurs in a band outside of hired spectrum ~\cite{Ref49}).
4. In terms of interference caused by PTs on secondary receivers, which mainly depends on distance between them, the aim of the secondary receiver is to be able to decode the primary signal only to help to accomplish a better secondary rate. However, it is not the scope of this paper to analyse this effect. In the literature, there are several studies on this topic. For instance, opportunistic interference cancellation (OIC) mechanism proposed in [34] can be utilized for this purpose by (a) selection of the data rate in the PT and (b) the link quality between the PT and the secondary receiver.
5. The proposed schemes aim to maximize multichannel allocation performance of hybrid CRNs with energy harvesting capability under interference constraint. STs need to utilize one or more channels to make transmission over and the allocation scheme means working best if it enables STs to be assigned all the channels they need. That' s why, the ChDistFairness factor given in Equation (9) is used to measure and prove the performance (fairness) of the proposed algorithms, compared to the greedy one. If STs get all channels they need, metric becomes 0 which means scheme makes perfect allocation and allocation performance is maximized. Consequently, abstract mentions that schemes proposed try to maximize allocation performance and all the results are obtained and compared in terms of this fairness metric.
6. The purpose of multi channel allocation is to assign channels such that all STs utilize channels they need. To measure the fairness, in other words effectiveness, of proposed schemes, Equation (9) is proposed in [42]. For this purpose, the same metric is used to prove the fairness of the proposed schemes in this journal too.
7. Prioritization of STs based on their energy level, whether they can harvest energy and number of channels they need are critical during distribution of channels to STs.
• Having higher initial energy enables an ST to sense channels and transmit data over longer time slots as their energy will be wasted in a longer period.
• If an ST is located in HZ of a PT, which means it can harvest energy, it will have more energy on the fly which means more chance to sense channels and make transmission.
• Having more than one wireless interface for transmission, meaning it has multi-channel usage ability, enables an ST to make transmission over more channels at the same time which improves throughput of the total system because aim of a cognitive system is to maximize channel utilization.
So from two nodes competing, the one with higher energy should obtain the channel. Nodes located in the HZ should be prioritized since harvesting capability contributes to have more energy and hence more ability to transmit data. Additionally, multi-channel transmission capability of an ST should have higher prioritization during channel allocation.
Considering these, this journal proposes a novel metric called PriorityMeasure given in Equation (6) to define the priority for each ST towards a channel j and one of the proposed algorithms is based on this metric for prioritizing STs to assign channels.
EL(i) is the current energy level of ST(i) in joules and max(EL) is maximum energy level of all STs. So, in the equation, component of energy level is normalized by maximum energy level.
CN(i) indicates the number of channels needed by ST(i), so, for example, if three channels are needed, the contribution to PriorityMeasure will be 0.9, if one channel, it will be 0.3. This component is appended to the metric, because, as mentioned before, contribution of STs owning higher number of channels on PriorityMeasure should be higher.
Finally, STh(i, j) is 1 if ST(i) is located in the HZ of the PT(j), and 0 otherwise. As an ST that can harvest should be located in guard zone by definition, this coefficient is separately added to the PriorityMeasure.
Coefficients 0.4, 0.3 and 0.3 are selected to prioritize and normalize each component such that sum of them becomes 1. Highest priority is given to the energy component as higher energy is most important factor for an ST to be able to make transmission.
Thank you for your invaluable comments. Best Regards.

Reviewer 3 Report
The paper addresses a current problem. The article is well documented and several simulations were conducted to support the conclusions. However, there are some minor corrections that are needed to improve the article.
Minor spell check is required. By example- In the Abstract is written „this paper proposes a novel ....algorithms”
- At page 2, is written „tarious” instead various
- The last sentence from Section 1, should be reformulated.
Please define all the notations and parameters in equations (2) and (7). Please explain why or how did you choose the constants 0.3 and 0.4 in Equation (6). For a better readability and understanding of the presented algorithms the author should write them in pseudocode. The selection statement IF and the WHILE loop, instead of the statment GO TO and the appropriate indenting should be used. The text in figure 3 is difficult to read. Increase the table width and font size. The next paragraph should be reformulated because it is ambiguous. What does „Here” represent? The equation (6) was not defined in [39].„To demonstrate the performance and supremacy of the proposed algorithms, the channel distribution performance for each of the mechanisms was compared. Here, the metric given in Equation (6) was used as the performance criterion, as defined previously [39], which also considers if any outage of PRs occurs due to exceeding the limit of interference threshold for the defined SINR.”
Author Response
Dear Reviewer,
Here are my replies:
The proposed allocation schemes are reorganized in tabular form so that they can be more readable and understandable. Also, the font size in the tables made a little smaller such that each algorithm fits to one page.
Note: Corrections and updates in the journal are highlighted by yellow too ("Please see the attachment as well").
Minor corrections:
The declaration in the abstract "this paper proposes a novel ....algorithms” has been corrected as "..algorithm".
At page 2, "tarious” has been replaced with "various".
The last sentence from Section 1 has been corrected as "Section 4 provides the concluding remarks of this paper."
All the notations and parameters in equations (2) and (7) are defined and explained in the journal as below:
Equation (2)
EL = P X T
where EL is energy level in Joule, P is power level in W and T is duration in seconds when power level P is used.
Equation (7)
• betaPR is the tolerable SINR ratio threshold at PR (SINR value when total interference is PUth which is the tolerable interference threshold at the primary receiver PR), so that the corresponding primary receiver can successfully receive primary transmitter packets;
• P_P_T is the transmission power of primary transmitter P_T;
• d_{(P_T,P_R )$ is the distance between primary transmitter P_T and primary receiver P_R;
• S is the total number of secondary users;
• PCi is the transmission power of secondary user si;
• d(Ci,PR) is the distance between si and the primary receiver PR; and
• No is additive white Gaussian noise for the receiver, with a zero mean.
The reason why coefficients 0.4, 0.3 and 0.3 have been selected has been explained in the journal. They are to prioritize and normalize each component in the equation such that sum of them becomes 1. Highest priority is given to the component representing energy level as having higher energy is the most important factor for an ST to be able to make transmission.
All algorithms have been reorganized in tabular one column form.
"If .. then.. else" statements have been used to show conditional blocks.
The size of figure 3 has been increased so that becomes more readable.
The next paragraph has been reorganized and corrected. "Here.." has been replaced by ".. in this journal". Equation (6) has been corrected as equation (9). "To demonstrate the performance and supremacy of the proposed algorithms, the channel distribution fairness (performance) for each of the mechanisms is compared. In order to show fairness of proposed allocation methods in this journal, the metric given in Equation (9) is used as the performance criterion [42], which also considers if any outage of PRs occurs due to exceeding the limit of interference threshold for the defined SINR."
Thank you for your invaluable comments. Best Regards.

Round 2
Reviewer 1 Report
I would like to thank the author for considering all my notes and comments. I have just one more note, figure 3 in page 9 needs to enlarge more. So, a reader can easily read it.